# Meningeal lymphatic supporting cells govern the formation and maintenance of zebrafish mural lymphatic endothelial cells

Xiang He [1,4], Daiqin Xiong [1,4], Lei Zhao[2], Jialong Fu[1] & Lingfei Luo [1,3] ✉

The meninges are critical for the brain functions, but the diversity of meningeal cell types and intercellular interactions have yet to be thoroughly examined. Here we identify a population of meningeal lymphatic supporting cells (mLSCs) in the zebrafish leptomeninges, which are specifically labeled by *ependymin*. Morphologically, mLSCs form membranous structures that enwrap the majority of leptomeningeal blood vessels and all the mural lymphatic endothelial cells (muLECs). Based on its unique cellular morphologies and transcriptional profile, mLSC is characterized as a unique cell type different from all the currently known meningeal cell types. Because of the formation of supportive structures and production of pro-lymphangiogenic factors, mLSCs not only promote muLEC development and maintain the dispersed distributions of muLECs in the leptomeninges, but also are required for muLEC regeneration after ablation. This study characterizes a newly identified cell type in leptomeninges, mLSC, which is required for muLEC development, maintenance, and regeneration.

The meninges provide mechanical support and protective barriers for the central nervous system (CNS) but also play indispensable roles in brain development, neuroinflammation, and CNS immunity[1–3]. Along the axis from the skull to the brain parenchyma, the meninges consist of the external dura mater attached to the skull, the arachnoid mater in the middle, and the inner pia mater covering the parenchyma[1–3]. The arachnoid, the subarachnoid space filled with cerebrospinal fluid, and the pia mater are referred to as the leptomeninges[1]. The leptomeninges of mammals are composed of fibroblasts, neural progenitor cells, immune cells, perivascular cells, and meningeal blood vessels connected to the parenchymal vasculature[4–6]. Although multiple meningeal cells have been known to be important for the proper functioning of the brain, the constitutions and functions of meningeal cell types remain to be more thoroughly understood.

A network of lymphatic vessels consisting of a single layer of blind-ended lymphatic endothelial cells (LECs) is distributed throughout most body tissues and is responsible for maintaining interstitial fluid homeostasis, immune surveillance, and lipid absorption[7]. The meningeal lymphatic vessels in the dura mater of mice drain cerebrospinal fluid, macromolecules, and immune cells from the CNS to the peripheral lymphatic network, thus participating in the regulation of immune surveillance and inflammatory responses in the brain[8–10]. The presence of meningeal lymphatics has also been found in the meninges of nonhuman primates and humans[11], and has recently been detected in the juvenile and adult zebrafish[12]. Furthermore, a population of perivascular LECs have been revealed to reside in the membranes attached to the zebrafish brain parenchyma[13–15], and named as mural lymphatic endothelial cells (muLECs)[15]. These muLECs are transcriptomically characterized as an LEC population and express canonical LEC markers including *prox1a*, *lyve1b*, *vegfr3*, and *mrc1a*, and muLEC development is dependent on the *vegfc/vegfr3/ccbe1* signaling axis[13–15]. Notably, under physiological conditions, muLECs do not form lymphatic vessels but consistently remain as dispersed single-cell communities[13–15]. The characteristics of non-tubular muLECs are

[1]Institute of Developmental Biology and Regenerative Medicine, Southwest University, Beibei, Chongqing 400715, China. [2]Shaanxi Key Laboratory of Qinling Ecological Intelligent Monitoring and Protection, School of Ecology and Environment, Northwestern Polytechnical University, Xi'an, Shaanxi 710072, China. [3]School of Life Sciences, Fudan University, Yangpu, Shanghai 200438, China. [4]These authors contributed equally: Xiang He, Daiqin Xiong. ✉e-mail: lluo@swu.edu.cn

distinct from those of tubularized peripheral and intracranial LECs. Apart from eliminating various metabolic wastes from the brain[14,16], muLECs, as a type of perivascular cells, have also been found to be involved in regulating the development of the meningeal vasculature[15] and assisting the post-injured cerebrovascular regeneration[17,18]. However, mechanisms underlying the development of muLECs as well as the formation and maintenance of their dispersed distributions in the leptomeninges remain unclear.

In this study, we identified a population of meningeal lymphatic supporting cells (mLSCs) specifically labeled by *ependymin* (*epd*) in zebrafish leptomeninges. These flat and irregular spindle-shaped mLSCs accumulate at 2 days post-fertilization (dpf) in a partially overlapping manner, forming a membranous structure that covers the brain and maintains throughout adulthood. mLSCs exhibit a unique transcriptomic profile and express unique markers that distinguish them from other known meningeal cells. muLECs always locate on the mLSC-composed membrane. Functionally, at early developmental stages, mLSCs provide migratory tracks and produce pro-lymphangiogenic factors to activate the germination of muLECs. Later, mLSCs are essential for maintaining the morphologies and dispersed distributions of non-tubular muLECs as well as muLEC regeneration after ablation. Collectively, these findings improve understanding of cellular constitutions and interactions of vertebrate meninges.

## Results

### *epd* is expressed in the larval zebrafish brain

Ependymin, encoded by gene *epd*, is a secretory glycoprotein first discovered in the extracellular fluid and cerebrospinal fluid of the goldfish brain[19]. According to previous reports showing specific expression of *epd* mRNA in the zebrafish leptomeninges[20], we examine whether *epd* is generally expressed in the whole leptomeninges or exclusively labeled a group of meningeal cells. Whole-mount in situ hybridization (WISH) on zebrafish larvae at 5 dpf showed that *epd* was exclusively expressed over the brain (Supplementary Fig. 1a), accumulated at the dorsal side of the brain mainly in symmetrical circles over the optic tectum (TeO) and in horizontal lines over the cerebellum (Supplementary Fig. 1a, b). This expression pattern of *epd* was reminiscent of muLECs, which similarly form bilateral circular structures on the TeO at this developmental stage[13–15]. However, fluorescence in situ hybridization (FISH) combined with antibody staining (FISH-antibody staining) in the *Tg(lyve1b:EGFP)* transgenic line[18,21] showed that *epd* was expressed around muLECs rather than in muLECs (Supplementary Fig. 1c). Furthermore, *epd* appeared to be around the tubular structures that were assumed as mesencephalic vein (MsV) (Supplementary Fig. 1c, arrowheads), which was confirmed by the FISH-antibody staining under the *Tg(kdrl:GFP)* transgenic background (Supplementary Fig. 1d, arrowheads).

### *epd*-positive cells enwrap meningeal blood vessels and muLECs

To further explore the nature of the *epd*-positive cells, we generated a *Tg(epd:EGFP)* transgenic line, in which EGFP was driven by the *epd* promoter. The distributions of EGFP-positive cells in the brain (Supplementary Fig. 1e–h) were highly similar to the location of *epd* mRNA expression, suggesting that the 3.9 kb of *epd* promoter could drive its endogenous expression. DAPI staining under the *Tg(epd:EGFP)* background at 5 dpf showed the nuclei of *epd*-positive cells as well as the adjacent nuclei of *epd*-negative cells (Supplementary Fig. 1i). In the *Tg(epd:EGFP; kdrl:mCherry-Ras)* and *Tg(epd:EGFP; lyve1b:DsRed)* zebrafish brains from larvae to adults, the *epd*-positive cells, although widely distributed on TeO, were more clustered around muLECs and meningeal blood vessels at multiple stages (Fig. 1a–f). Higher magnification images revealed the presence of visible cavities of muLECs/blood vessels among *epd*-positive cells

(Fig. 1a–d, arrowheads), implicating that the *epd*-positive cells were different from either muLECs or meningeal blood vessels.

Detailed examination of the *epd*-positive cells, muLECs, and meningeal vasculature on the surface of the brains of 5-dpf larval and 7-mm juvenile zebrafish revealed the constant presence of *epd*-positive cells around muLECs (Fig. 1a–d), but the blood vessels were not always around muLECs (Fig. 1a'–d', and Supplementary Fig. 2a–d). These results were further validated by the intracerebroventricular (ICV) injection of fluorescent macromolecule Alexa647-IgG (150 kDa) into the *Tg(epd:mCherry-NTR; fli1:GFP)* transgenic larvae (Supplementary Fig. 2e–g). Although *fli1* labels both muLECs and blood vessel endothelial cells (BECs), only muLECs can take up Alexa647-IgG[14]. Thus, BECs were only labeled by GFP, while muLECs were double-labeled by GFP and Alexa647-IgG. In some brain areas in particular the posterior half of the midbrain and the midbrain-hindbrain boundary, muLECs were always enwrapped by the *epd*-positive cells, but not adjacent to blood vessels (Supplementary Fig. 2f–h). These data suggest that muLECs are more closely associated with *epd*-positive cells than with blood vessels during development.

### *epd*-positive cells present in zebrafish leptomeninges

In adults at 4 months post-fertilization (mpf), the *epd*-positive cells were shown to be absent in the dura mater attached to the skull (Fig. 1g), whereas were widely distributed in the midbrain and hindbrain and still featured by encompassing blood vessels and muLECs (Fig. 1e, f). Sagittal sections of the adult zebrafish brain showed that the *epd*-positive cells were localized in the olfactory bulb, the base of the forebrain, midbrain, and hindbrain, the regions where muLECs settled, and were absent in the brain parenchyma (Fig. 1i, k). Cross-sections of the adult brain demonstrated that the *epd*-positive cells were only detected in the meninges and further illustrated that the meningeal vasculature and muLECs were embedded in the membranes formed by the *epd*-positive cells (Fig. 1j, l). Altogether, these meningeal *epd*-positive cells are exclusively resident in the leptomeninges throughout zebrafish lifespan and consistently around muLECs and most of meningeal blood vessels.

### *epd*-positive cells are not other known meningeal cell types

The mammalian leptomeninges contain a variety of cell types, including fibroblasts, neural progenitor cells, LECs, BECs, perivascular cells, and various immune cells[1,4–6,22–24]. To investigate whether the *epd*-positive cell belongs to one of the common meningeal cell types, we first analyzed the brains of *Tg(epd:H2B-mCherry; fli1:nEGFP)* zebrafish at 5 dpf and 4 mpf. Co-localization between the H2B-mCherry-labeled nuclei and nEGFP-labeled nuclei was never observed (Fig. 2a, b), further indicating that the *epd*-positive cell did not belong to endothelial cell types. The fibroblasts labeled by Collagen I are the main component of mouse brain membranes[6]. The sections of adult zebrafish brain at 6 mpf displayed that the Collagen I-labeled fibroblasts located at a layer different from, but immediately beneath, the *epd*-positive cell layer (Fig. 2c), suggesting that the *epd*-positive cells are not fibroblasts, but form a membranous structure similar to fibroblasts. In multiple transgenic lines including *Tg(abcc9*BAC*:Gal4ff; UAS:GFP)* labeling pericytes[25], *Tg(pdgfrb*BAC*:EGFP)* labeling pericytes and/or vascular smooth muscle cells[26], *Tg(acta2:GFP)* labeling vascular smooth muscle cells[27], *Tg(coro1a:Kaede)* labeling macrophages and neutrophils[28], *Tg(lyz:GFP)* labeling neutrophils[29], *Tg(mpeg1:GFP)* labeling macrophages[30], *Tg(prox1a*BAC*:KalTA4; UAS:TagRFP)* labeling LECs and neuronal cells[31], *Tg(nkx2.2a:GFP)* labeling neuronal and/or oligodendrocyte cells[32], and *Tg(elavl3:GFP)* labeling neurons[33], the *epd*-positive cells did not share any overlap or morphological similarities with these cell types (Fig. 2d–l). Hence, the *epd*-positive cell is anatomically distinct from the currently known meningeal cell types.

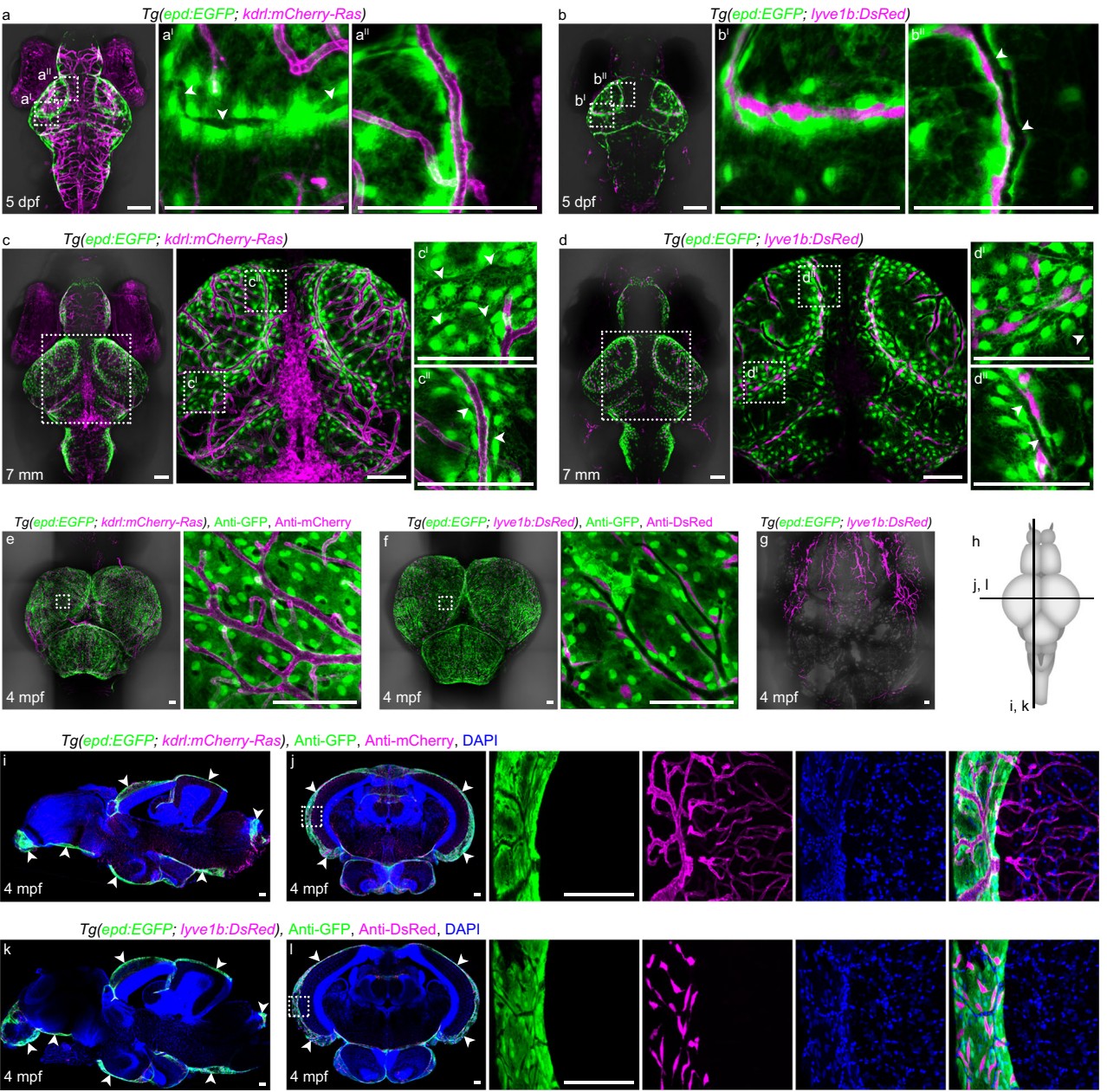

**Fig. 1 | The *epd*-positive cells in zebrafish leptomeninges consistently enwrap muLECs from larva to adult. a**, **c**, **e** Dorsal confocal images of *epd*-positive cells and blood vessels on *Tg(epd:EGFP; kdrl:mCherry-Ras)* brains at 5 dpf (**a** *n* = 20), 7 mm (**c** *n* = 20) and 4 mpf (**e** *n* = 12). White arrowheads indicate the cavities of mural lymphatic endothelial cells (muLECs). **b**, **d**, **f** Dorsal confocal images of *epd*-positive cells and muLECs on *Tg(epd:EGFP; lyve1b:DsRed)* brains at 5 dpf (**b** *n* = 20), 7 mm (**d** *n* = 20) and 4 mpf (**f** *n* = 12). White arrowheads indicate the cavities of meningeal blood vessels. **g** A representative confocal image of the meningeal lymphatics in the dura mater beneath the skull at 4 mpf. *n* = 12. **h** Illustration of cross-sections (horizontal line) and sagittal sections (vertical line) of adult zebrafish brain in (**i–l**). **i–l** Confocal images of brain sections showing the locations of *epd*-positive cells in the leptomeninges at 4 mpf. White arrowheads indicate the distributions of *epd*-positive cells. *n* = 12 adult brains per experiment. Each experiment was repeated three times independently with similar results. The white dashed boxes outline the enlarged areas. Scale bars: 100 μm.

The previously reported single-cell RNA-sequencing (scRNAseq) data from whole zebrafish embryos and larvae as well as adult telencephalon revealed that *epd* is specifically expressed in a group of cells[34–36]. Here, we re-analyzed the scRNAseq data from whole zebrafish embryos and larvae by Farnsworth et al.[34]. On the basis of marker genes and other zebrafish scRNAseq results[1,4,6,23,34,35], we picked out *epd*-positive cells and common meningeal cell types (including fibroblasts, neural progenitor cells, endothelial cells, mural cells, and immune cells) for analyses (Supplementary Data 1). We found that each cluster was featured by distinctive gene expression patterns (Supplementary Fig. 3a–c). Particularly, the *epd*-positive cells possessed their

exclusively highly expressed genes that have previously not been reported as marker genes for other meningeal cell populations (Supplementary Fig. 3d). Taken together, these data suggest that *epd*-positive cells represent a meningeal cell type with unique anatomical and transcriptomic characteristics.

### *epd*-positive cells produce pro-lymphangiogenic factors

To further understand the molecular characteristics of the early-stage *epd*-positive cells, we isolated the *epd*-positive cells from zebrafish brains at 55 h post-fertilization (hpf) and 5 dpf by fluorescence-activated cell sorting (FACS) (Fig. 3a, b) and carried out transcriptomic

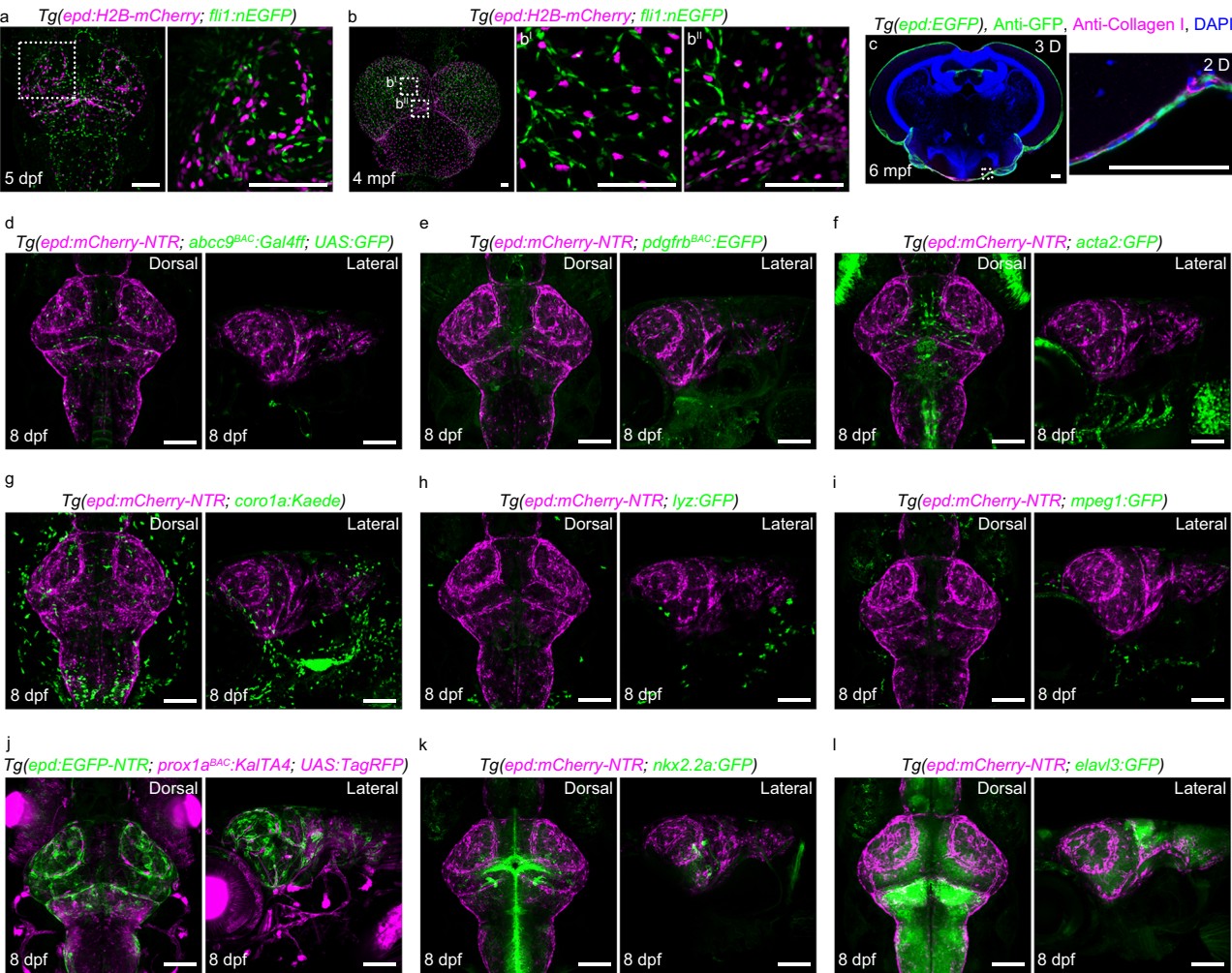

**Fig. 2 | The *epd*-positive cells do not belong to other known meningeal cell types. a, b** Dorsal confocal images of *epd:H2B-mCherry*-positive nuclei and *fli1:nEFP*-positive nuclei in *Tg(epd:H2B-mCherry; fli1:nEFP)* brains at 5 dpf (**a** n = 20) and 4 mpf (**b** n = 8). **c** Confocal image of a cross-section of an adult brain showing collagen I-labeled fibroblasts at a different layer to *epd*-positive cells at 6 mpf. A 2D view of the enlarged area is shown. n = 8 adult brains. The experiment was repeated three times independently with similar results. **d–l** Confocal images showed *epd*-positive cells did not co-stain with *abcc9^{BAC}:Gal4ff; UAS:GFP*-positive pericytes (**d**),

*pdgfrb^{BAC}:EGFP*-positive pericytes (**e**), *acta2:GFP*-positive smooth muscle cells (**f**), *coro1a:Kaede*-positive immune cells (**g**), *lyz:GFP*-positive neutrophils (**h**), *mpeg1:GFP*-positive macrophages (**i**), *prox1a^{BAC}:KalTA4; UAS:TagRFP*-positive lymphatic endothelial cells and neuronal cells (**j**), *nkx2.2a:GFP*-positive neuronal and/or oligodendrocytes (**k**), and *elavl3:GFP*-positive neurons (**l**) at 8 dpf. Dorsal and lateral views of the larval brains were shown. n = 20 per experiment. Each experiment was repeated three times independently with similar results. The white dashed boxes outline the enlarged areas. Scale bars: 100 μm.

profiling by RNA-seq analyses. To efficiently separate the *epd*-positive cells from adjacent muLECs, FACS was performed using the *Tg(epd:EGFP-NTR; lyve1b:DsRed)* transgenic line (Fig. 3c, d). The sorted *epd*:EGFP+ cells were applied for RNA-sequencing analyses, using whole embryos/larvae at 55 hpf and 5 dpf as controls. Principal component analysis verified the similarities between our RNA-seq data and the published scRNAseq data[34] of *epd*-positive cells (Supplementary Fig. 3e). Then, the expressions of various meningeal cell type marker genes were analyzed in the sorted *epd*-positive cells and in whole fish. The makers of immune cells (ImCs), glial and neural cells (GAN), BECs, pericytes (PCs), LECs, and fibroblasts (FBs) failed to express or expressed at very low levels in the *epd*-positive cells (Fig. 3e; Supplementary Data 2). By contrast, the *epd*-positive cells expressed high levels of their unique markers such as *slc13a4, soul5, slc7a2, igfbp2a, nid1b, apof, ggctb*, and *epd* (Fig. 3e; Supplementary Data 2), which were validated by WISH and FISH-antibody staining (Supplementary Fig. 4).

Notably, we revealed that although the *epd*-positive cells did not express neurotrophic factors or pro-angiogenic factors, they expressed high levels of pro-lymphangiogenic factors such as *vegfc, vegfd,*

*ccbe1* and *mmp2* (Fig. 3f). Analysis of Enriched Ontology Clusters for highly expressed genes in the *epd*-positive cells relative to whole fish showed that the *epd*-positive cell-enriched genes have terms associated with the SLC-mediated transmembrane transport, import into the cell, mesenchyme development, and lymphangiogenesis (Supplementary Fig. 5; Supplementary Data 3, 4). Thus, the *epd*-positive cells represent a previously unidentified leptomeningeal cell population that produce pro-lymphangiogenic factors.

## A portion of muLECs migrate along *epd*-positive cells

Since the *epd*-positive cells produce pro-lymphangiogenic factors, we first track their development to see whether they develop prior to the muLEC formation. Live imaging of zebrafish embryos from 0 hpf was carried out. The appearance of *epd*-positive cells was first detected in the midbrain at the 18-somite-stage (ss) (Fig. 4a), and then the number of this cell population increased from 18 ss to 24 ss (Supplementary Movie 1). To display the morphologies of individual *epd*-positive cell, we generated the *Tg(epd:H2B-GFP; epd:mCherry-Ras)* transgenic strains, in which the membranes and nuclei of *epd*-positive cells were

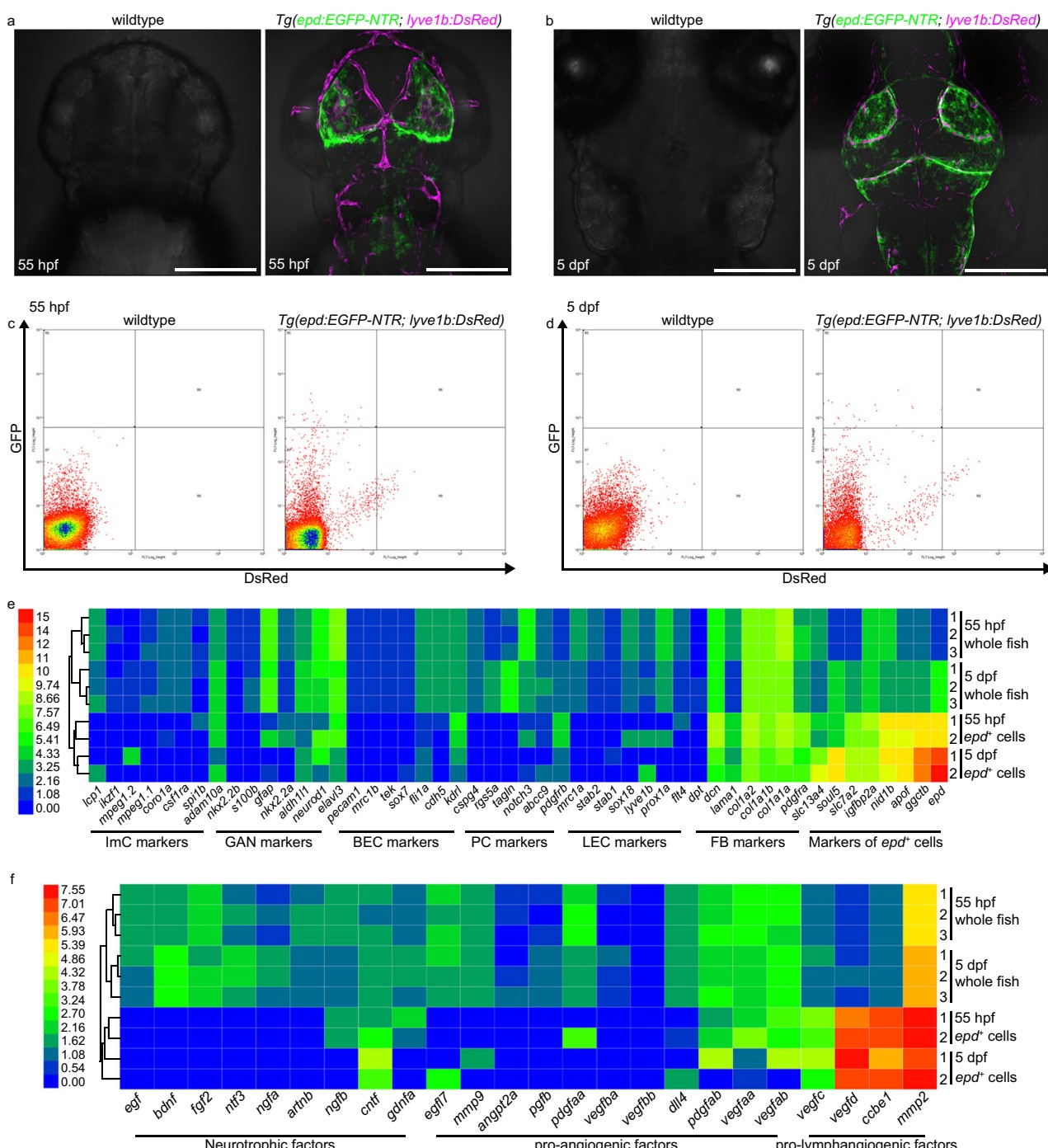

**Fig. 3 | The *epd*-positive cells express unique markers and produce pro-lymphangiogenic factors at embryonic/larval stages. a, b** Dorsal confocal images of fluorescence-negative wild-type and *Tg(epd:EGFP-NTR; lyve1b:DsRed)* embryonic/larval brains at 55 hpf (**a**) and 5 dpf (**b**). *n* = 30 embryos/larvae per panel. Scale bars: 200 μm. **c, d** Representative plots for fluorescence-activated cell (FAC) sorted EGFP-NTR-positive cells in brains of fluorescence-negative wild-type and *Tg(epd:EGFP-NTR; lyve1b:DsRed)* at 55 hpf (**c**) and 5 dpf (**d**). **e** Heatmap for differential expression levels of *epd*-positive cells and other selected known meningeal cell marker genes between whole fish (*n* = 100 fish per replicate) and FAC sorted

*epd*-positive cells (EGFP-NTR-positive, *n* = 100 cells per replicate). ImC immune cell, GAN glial and neural cells, BEC blood vessel endothelial cell, PC pericytes, LEC lymphatic endothelial cell, FB fibroblasts, *epd*+ cells *epd*-positive cells. Scale bar represents relative expression by log2(FPKM + 1), from 0 (lowest) to 15 (highest). **f** Heatmap for differential expression levels of neurotrophic, pro-angiogenic, and pro-lymphangiogenic factors between whole fish (*n* = 100 fish per replicate) and FAC sorted *epd*-positive cells (EGFP-NTR-positive, *n* = 100 cells per replicate). Scale bar represents relative expression by log2(FPKM + 1), from 0 (lowest) to 7.55 (highest).

labeled by the membrane-targeted mCherry-Ras and the nuclear H2B-GFP, respectively (Fig. 4b). The *epd*-positive cells formed a membrane structure covering the TeO from 2 dpf (Fig. 4b). At 5 dpf, individual *epd*-positive cell in forebrain, midbrain, and hindbrain all exhibited a flattened irregular spindle cell shape, and multiple cells look like

partially stacked rather than interconnected together in terms of the overlap between cell membranes (Fig. 4b, Supplementary Fig. 6a, b).

The muLECs originate in the optic choroidal vascular plexus (OCVP), then posteriorly migrate either along the medial side of MsV or along the ventral side of the midbrain, finally forming a classical

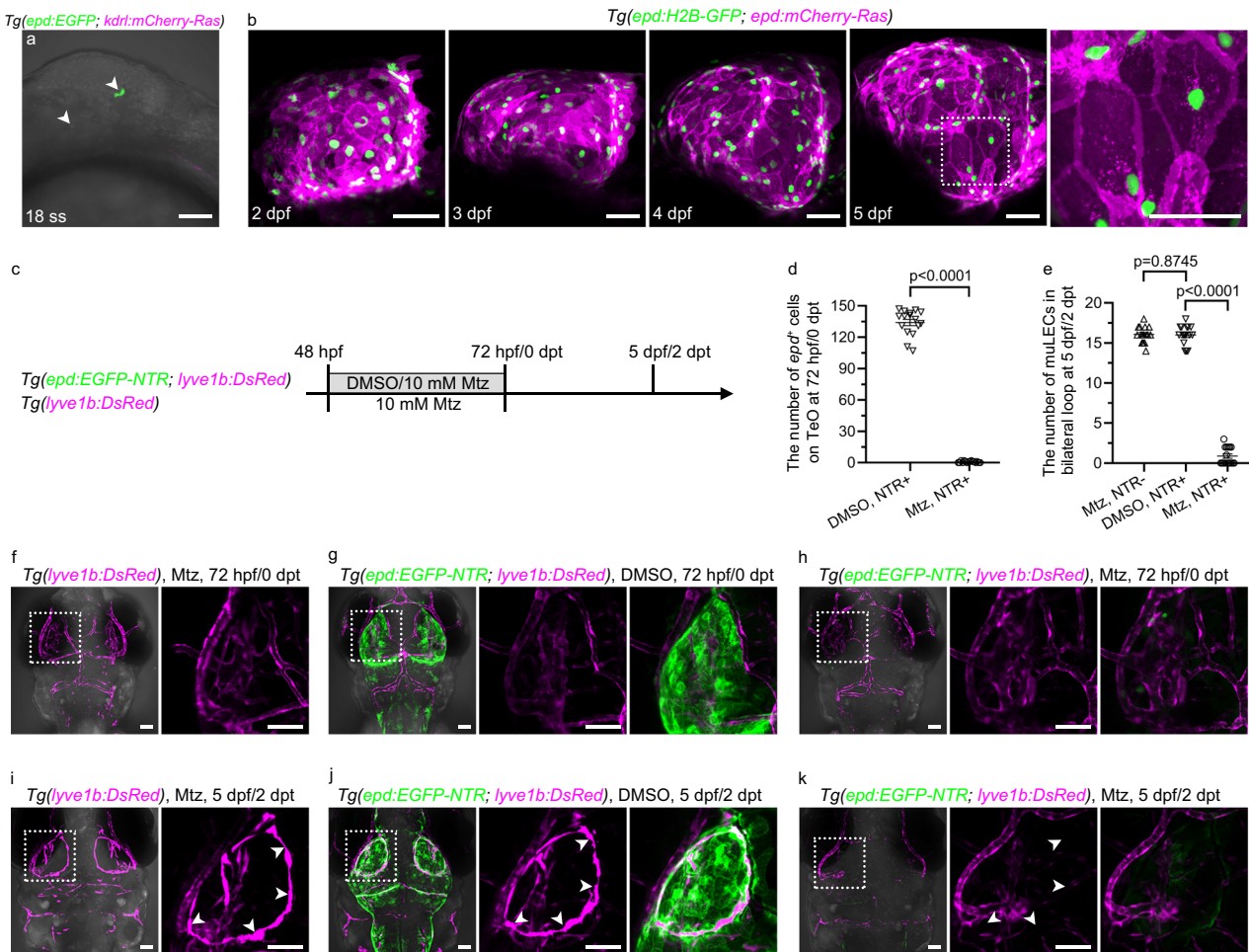

**Fig. 4 | The *epd*-positive cells are required for the development of early muLECs. a** A representative lateral confocal image of *epd*-positive cells first appeared in *Tg(epd:EGFP; kdrl:mCherry-Ras)* brains at 18-somite stage (ss). White arrowheads indicate the *epd*-positive cells. *n* = 10. The experiment was repeated three times independently with similar results. **b** Lateral confocal images of the development of *epd*-positive cells in *Tg(epd:H2B-GFP; epd:mCherry-Ras)* from 2 dpf to 5 dpf. The nuclei and membranes of *epd*-positive cells are labeled with H2B-GFP and mCherry-Ras, respectively. Enlarged area showing irregular spindle-shaped *epd*-positive single-cell. *n* = 15 per stage. **c** Schematic diagram showing the strategy for ablation of *epd*-positive cells. **d**, **e** Quantification of the number of *epd*-positive cells on the optic tectum (TeO) at 72 hpf/0 dpt (**d**) and the number of mural lymphatic endothelial cells (muLECs) in bilateral loop at 5 dpf/2 dpt (**e**) in the non-ablation and *epd*-positive cell ablation group. 16 fish were observed in three

independent experiments in each group. **f, g, i, j** Dorsal confocal images of *epd*-positive cells and muLECs in brains that did not induce muLEC or *epd*-positive cell injury at 72 hpf/0 dpt and 5 dpf/2 dpt. White arrowheads indicate muLECs. *n* = 34 per experiment. **h** Dorsal confocal images of ablation of *epd*-positive cells in brains at 72 hpf/0 dpt. *n* = 38. The experiment was repeated three times independently with similar results. **k** Dorsal confocal images of massive missing in muLECs after *epd*-positive cell ablation at 5 dpf/2 dpt. White arrowheads indicate where the muLECs should have been. *n* = 33. The experiment was repeated three times independently with similar results. Error bars, mean ± SEM. Unpaired two-tailed Student's *t*-test. *P* values included in the graphs. Source data are provided as a Source Data file. The white dashed boxes outline the enlarged areas. Scale bars: 50 μm.

bilateral symmetrical circular structure on the TeO at around 4 dpf[13,14]. Given the close proximity of *epd*-positive cells and muLECs, we next study whether muLECs migrate along the *epd*-positive cells during development. Time-lapse imaging from 60 hpf when muLECs begin to emerge to 106 hpf when typical muLEC loop structures are formed was carried out under the *Tg(epd:mCherry-NTR; lyve1b:EGFP)* transgenic background. Before the budding of muLECs from OCVP, the *epd*-positive cells already clustered (Supplementary Movie 2). Migration of muLECs along the ventral side of the midbrain turns out to follow a track provided by the *epd*-positive cells (Supplementary Movie 2, yellow arrowhead). The lateral framework of *epd*-positive cells at the midbrain-hindbrain junction formed at 60 hpf, and then muLECs sprouted out at about 77 hpf, again demonstrating guidance of muLEC migration by the *epd*-positive cells (Supplementary Movie 2, blue arrowhead). These results suggest that the *epd*-positive cells are potentially important for the development of muLECs.

## *epd*-positive cells regulate the development of muLECs

In order to explore whether the *epd*-positive cells are required for the formation of muLECs, we generated the *Tg(epd:EGFP-NTR)* and *Tg(epd:mCherry-NTR)* transgenic lines to ablate the *epd*-positive cells based on the nitroreductase-metronidazole (NTR-Mtz) system[37,38]. Because muLECs sprout from OCVP at approximately 54–56 hpf, treatment with Mtz was applied to the *Tg(epd:EGFP-NTR; lyve1b:DsRed)* transgenic lines from 48 hpf to 72 hpf, which resulted in the ablation of *epd*-positive cells (Fig. 4c, d, h). The *Tg(lyve1b:DsRed)* line treated with Mtz (Fig. 4c, f) and the *Tg(epd:EGFP-NTR; lyve1b:DsRed)* line treated with DMSO (Fig. 4c, g) were used as controls. At 2 days post Mtz treatment (dpt), equivalent to 5 dpf, the muLECs were massively missing, including the typical bilateral circular structures (Fig. 4e, i–k). These results were validated by using *prox1a* to label LECs under the *Tg(epd:EGFP-NTR; prox1a^BAC^:KalTA4; UAS:TagRFP)* transgenic background

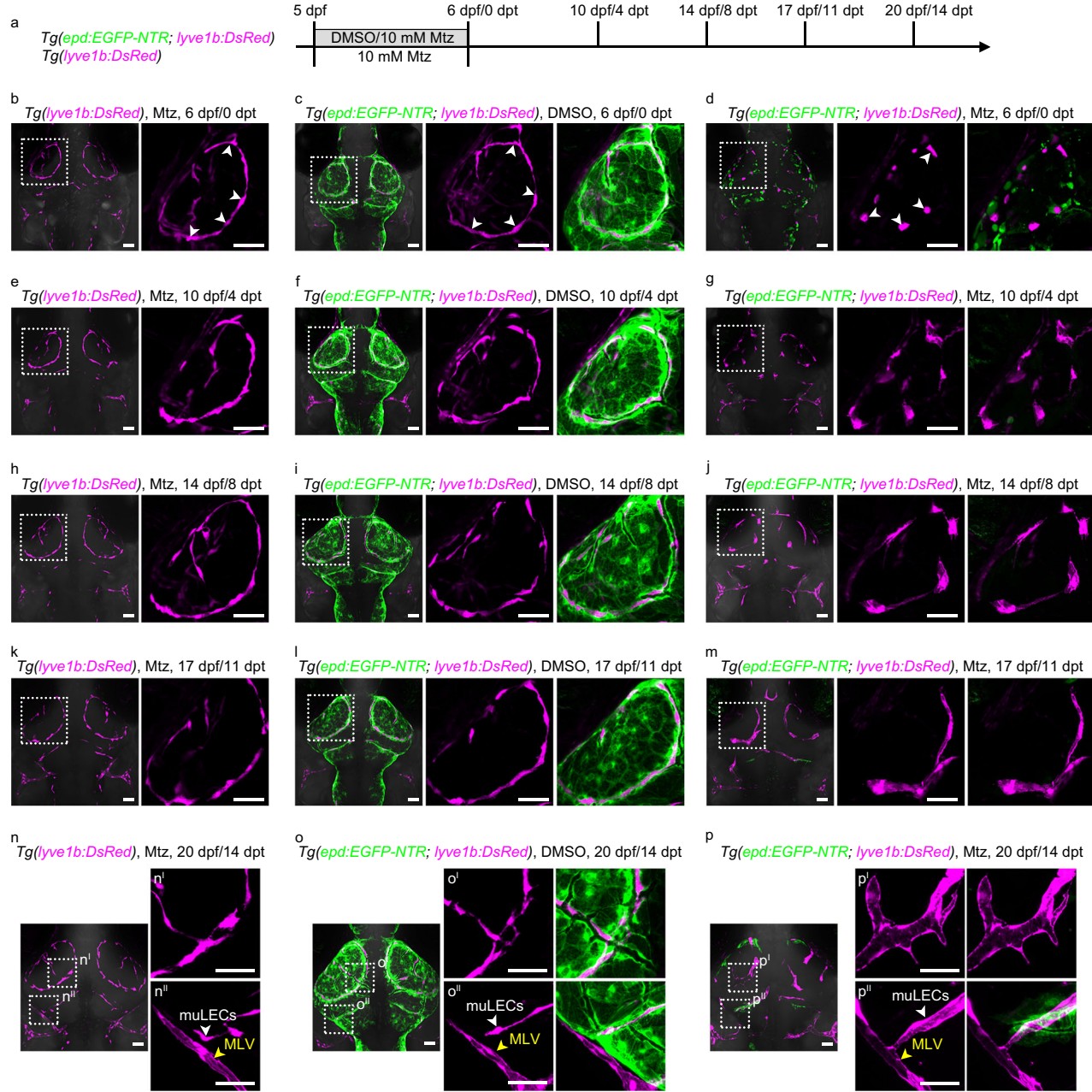

**Fig. 5 | The mLSCs are essential for maintaining the dispersed distribution pattern of muLECs at the larval stage. a** Schematic diagram showing experimental design for meningeal lymphatic supporting cell (mLSC) ablation. **b, c, e, f, h, i, k, l, n, o** Dorsal confocal images of mural lymphatic endothelial cells (muLECs) and mLSCs in brains that did not induce muLEC or mLSC injury at 6 dpf/ 0 dpt, 10 dpf/4 dpt, 14 dpf/8 dpt, 17 dpf/11 dpt and 20 dpf/14 dpt. White arrowheads indicate muLECs. Yellow arrowheads indicate meningeal lymphatic vessels (MLV).

*n* = 20 per experiment. **d, g, j, m, p** Dorsal confocal images showing the progressive formation of lymphatic vessels from collapsed muLECs after inducing injury to mLSCs. White arrowheads in (**d**) indicate collapsed muLECs. White arrowheads in (**p**) indicate muLEC-derived lymphatic vessels. Yellow arrowheads indicate MLV. **d** *n* = 28; **g** *n* = 24; **j** *n* = 20; **m** *n* = 17; **p** *n* = 19. The experiments were repeated three times independently with similar results. The white dashed boxes outline the enlarged areas. Scale bars: 50 μm.

(Supplementary Fig. 6c–e). Using the Alexa647-IgG macromolecule endocytosis function to label muLECs, intracranial injection of Alexa647-IgG further confirmed the loss of muLECs after ablation of the *epd*-positive cells (Supplementary Fig. 6f–h). Time-lapse imaging from 60 hpf to 106 hpf using the *Tg(epd:mCherry-NTR; lyve1b:EGFP)* transgenic lines revealed that after ablation of the *epd*-positive cells, muLECs failed to sprout from OCVP (Supplementary Movie 3). Taken together, all these data demonstrate that the *epd*-positive cell is an essential supportive cell type for muLEC sprouting and development, thereafter we name it as a meningeal lymphatic supporting cell (mLSC).

## Ablation of mLSCs causes muLECs to form lymphatic vessels

Although muLECs express multiple LEC markers, they do not form lymphatic vessels but maintain dispersed cell distributions[13–15]. Next, we examined whether mLSCs were responsible for the maintenance of muLECs at different stages. Mtz was applied to the *Tg(epd:EGFP-NTR; lyve1b:DsRed)* transgenic larvae to ablate mLSCs at 5 dpf, when the basic pattern of muLECs was initially established (Fig. 5a). At 6 dpf/0 dpt when Mtz was withdrawn after a 24-h-treatment, mLSCs were successfully ablated and the continuous endothelial loops of muLECs (Fig. 5b, c, Supplementary Fig. 7a) began to be fractured (Fig. 5d). Time-lapse live imaging of the Mtz-treated larvae from

5 dpf to 6.5 dpf illustrated that muLECs collapsed from elongated strips to ovals along with the ablation of mLSCs (Supplementary Movie 4). Although the morphologies of muLECs changed after mLSC ablation, the number of muLECs was maintained (Supplementary Fig. 7b). Moreover, TUNEL assays showed that muLECs did not undergo apoptosis (Supplementary Fig. 7c, d). After mLSC ablation, the adjacent muLECs converged to form cell clusters at 10 dpf/4 dpt (Fig. 5e–g), then protruded filopodium-like structures to interconnect at 14 dpf/8 dpt (Fig. 5h–j) and formed tubular-like structures at 17 dpf/11 dpt (Fig. 5k–m) (Supplementary Fig. 8a–d). Eventually, these muLECs became lumenized vessels at 20 dpf/ 14 dpt after mLSC ablation (Fig. 5n–p, Supplementary Fig. 8a–d).

Then, we analyze whether the loss of mLSCs leads to the formation of functional lymphatic vessels by muLECs. At 14 dpt after Mtz treatment, some muLEC-derived lymphatic vessels were observed to connect to the developing meningeal lymphatic vessels (MLVs) sprouting from otolithic lymphatic vessels (Fig. 5p^II, Supplementary Fig. 8e). MLVs play important roles in the immune surveillance of the brain and drainage of cerebrospinal fluid[8–10,12]. Time-lapse imaging under the *Tg(epd:EGFP-NTR; lyz:GFP; lyve1b:DsRed)* background at 20 dpf/14 dpt after mLSC ablation showed that the *lyz*:GFP+ lymphocytes slowly moved in the muLEC-derived lymphatic vessels (Fig. 6b, c). The muLECs in the physiological state at the same time do not carry *lyz*:GFP+ lymphocytes (Fig. 6a, c). The ICV injection of Alexa647-Dextran (10 kDa) into the brain of *Tg(epd:EGFP-NTR; lyve1b:DsRed)* juvenile at 20 dpf/14 dpt, which was capable of tracing cerebrospinal fluid (Fig. 6d), showed that the Alexa647-Dextran-labeled fluid filled the muLEC-derived vessels after mLSC ablation (Fig. 6h, n), in contrast to the muLECs under physiological conditions that endocytosed the dye (Fig. 6f, g, m). These results indicate that the mLSC ablation-induced, muLEC-derived lymphatic vessels obtain the functions of immune cell transport and cerebrospinal fluid drainage. Furthermore, the injection of Alexa647-Dextran into the dorsal aorta (DA) to image blood circulation (Fig. 6e) revealed that the dye was absent from the muLEC-derived vessels after mLSC ablation, suggesting separation of the muLEC-derived lymphatic vessels from blood vessels (Fig. 6i–k, m, n).

### mLSC ablation does not change vascular and pericyte morphology

Since mLSCs are also in close proximity to meningeal vessels, we therefore explored whether mLSC ablation affects the morphology of cerebral vasculature. After application of Mtz to the *Tg(epd:mCherry-NTR; kdrl:GFP)* larvae to ablate mLSCs, neither obvious alterations of MsV morphologies and cell number at 7 dpf/1 dpt nor brain BEC apoptosis at 6 dpf/0 dpt were observed (Supplementary Fig. 9a–f). Additionally, pericytes also remained normal at 7 dpf/1 dpt (Supplementary Fig. 9g, h). These results indicate that mLSCs are specific for the maintenance of muLECs, but not for brain vasculature.

### mLSCs maintain muLEC morphology in juvenile zebrafish

The studies above ablate the mLSCs in zebrafish larvae at 5 dpf, we next examine the phenomenon in juvenile zebrafish at 25 dpf (Supplementary Fig. 10a). Again, the mLSC ablation-induced muLEC morphological change, lymphatic vessel formation and connection to MLVs were observed (Supplementary Fig. 10b–e), indicating that the maintenance of muLEC morphologies and distributions by mLSCs is conserved among different stages. Taken together, all these results demonstrate that the cell morphologies and dispersed distribution pattern of muLECs are supported by mLSCs, and loss of mLSCs will induce formation of authentic functional leptomeningeal lymphatic vessels by muLECs.

### *sox10*-positive cell ablation does not alter muLEC morphology

To ensure the morphological changes of muLECs are specifically caused by ablation of mLSCs but not generally caused by massive cell death, we ablate oligodendrocytes and oligodendrocyte progenitor cells using the *sox10:EGFP-NTR* transgene to induce massive cell death in the brain. *sox10* labels oligodendrocytes and oligodendrocyte progenitor cells in zebrafish[39,40]. Mtz was applied to the *Tg(sox10:EGFP-NTR; lyve1b:DsRed)* transgenic larvae at 5 dpf (Supplementary Fig. 11a). At 6 dpf/0 dpt after Mtz treatment, the *sox10*-positive cells were successfully ablated. However, the morphologies of muLECs remained unaffected at 6 dpf/0 dpt, 8 dpf/2 dpt, and 10 dpf/4 dpt (Supplementary Fig. 11b–g). These results suggest that the morphological change of muLECs after mLSC ablation is not an indirect effect of massive cell death.

### mLSCs are required for muLEC regeneration

The muLECs are able to self-repair after partial injury by high-energy lasers[15]. Because of the pivotal roles of mLSCs in muLEC development and maintenance, the question whether mLSCs are involved in muLEC regeneration was investigated. The muLECs develop a lymphatic endothelial bilateral loop on TeO at 5 dpf (Fig. 1b), we chose this time point to induce muLEC ablation (Fig. 7a). Controlled by the *Tg(lyve1b:DsRed)* larvae treated with Mtz (Fig. 7b) and the *Tg(epd:mCherry-Ras; lyve1b:EGFP-NTR)* larvae treated with DMSO (Fig. 7c), application of Mtz to *Tg(epd:mCherry-Ras; lyve1b:EGFP-NTR)* larvae caused massive ablation of muLECs (Fig. 7d, o), which became partially regenerate at 11 dpf/5 dpt (Fig. 7e–g, p, q). By contrast, double ablation of muLECs and mLSCs could be successfully induced in the *Tg(epd:mCherry-NTR; lyve1b:EGFP-NTR)* transgenic larvae (Fig. 7h–k, o, r). But most muLECs failed to regenerate at 11 dpf/5 dpt (Fig. 7l–n, q). These evidence suggest that mLSCs are required for muLEC regeneration.

## Discussion

Our study reveals a previously unidentified population of lymphatic support cells in the leptomeninges of zebrafish that are required for the development, maintenance, and regeneration of muLECs. Fibroblasts are known to provide microenvironmental cues and mechanical support to surrounding cells in multiple organs[41]. In zebrafish, mLSCs form interconnected membranous structures and exhibit similar cell morphologies and functions to fibroblasts. Previous scRNAseq data classified mLSCs as a subclass of mesenchyme[35], and our RNA-sequencing analysis showed that the mLSC-enriched genes at 55 hpf also have a GO term of mesenchyme development, suggesting that mLSCs may be functional in supporting adjacent cells. Solute carriers (SLCs) mediate membrane transport of various substances and maintain the stability of the intracellular environment, and some of the SLCs have been identified to express in brain barrier cells[42]. The mLSC-enriched genes were shown to correlate with SLC-mediated transmembrane transport pathways, suggesting that mLSCs may carry out barrier functions in maintaining brain homeostasis.

The muLECs remain dispersed and relatively stationary throughout the zebrafish lifespan. They are invariably encapsulated by mLSCs and do not always coexist with meningeal blood vessels, suggesting a close association between muLECs and mLSCs. Previous studies have suggested that the early development of muLECs may migrate along blood vessels[13,14]. Based on our study, those muLECs migrating ventrally along the hindbrain and laterally along the midbrain-hindbrain junction where the meningeal blood vessel is absent should be directed and supported by mLSCs rather than by blood vessels. However, on the tracks of those muLECs migrating along MsV, both blood vessels and mLSCs are present. In these regions, the possibility that mLSCs and blood vessels cooperate to guide and support muLEC migration should not be excluded. The fundamental physiological function of muLECs under dispersed morphological conditions is their capacity to endocytose various types of substrates[14,16]. Previous studies have demonstrated distributions of tasks between muLECs and microglia in the endocytosis of extracellular cargo molecules from the brain[16]. After mLSC ablation, muLECs transform to vessels and no longer possess the

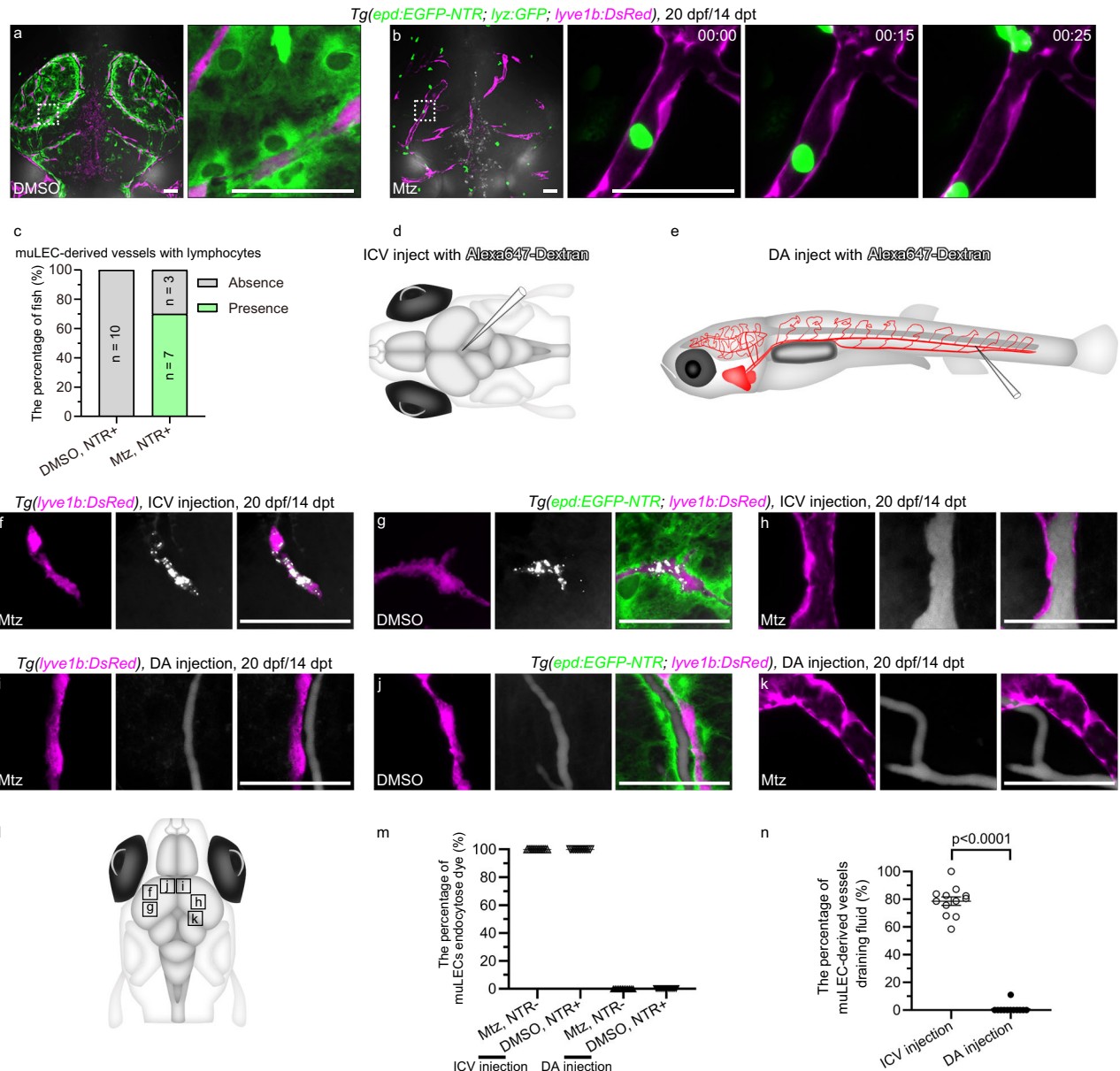

**Fig. 6 | The muLEC-derived lymphatic vessels perform the functions of transporting immune cells and draining cerebrospinal fluid. a** Dorsal confocal images of *lyz*:GFP+ lymphocytes in DMSO-treated brains at 20 dpf/14 dpt. *n* = 6. **b** Time-lapse images of the flow of *lyz*:GFP+ lymphocytes in mural lymphatic endothelial cell (muLEC)-derived lymphatic vessels of Mtz-treated brains at 20 dpf/14 dpt. The duration of time-lapse imaging is represented in hours:minutes. *n* = 6. The experiment was repeated three times independently with similar results. **c** Percentage of fish whose muLEC-derived vessels with (green) and without (gray) immune cells. 10 fish were observed in three independent experiments in each group. **d**, **e**, **l** Illustrations of intracerebroventricular (ICV) and dorsal aorta (DA) injection points of Alexa647-dextran and image areas. **f**, **g** Confocal images of the uptake of Alexa647-dextran by muLECs in meningeal lymphatic supporting cell (mLSC)-uninjured brains after ICV injection. *n* = 20 per experiment. Each experiment was repeated three times independently with similar results. **h** Confocal images of the uptake of Alexa647-dextran by muLEC-derived lymphatic vessels in mLSC-injured brains after ICV injection. *n* = 17. The experiment was repeated three times independently with similar results. **i**, **j** Confocal images of the uptake of Alexa647-dextran by blood vessels in mLSC-uninjured brains after DA injection. *n* = 14 per group. Each experiment was repeated three times independently with similar results. **k** Confocal images of the flow of Alexa647-dextran in blood vessels but not muLEC-derived lymphatic vessels in mLSC-injured brains after DA injection. *n* = 14. The experiment was repeated three times independently with similar results. **m**, **n** Quantification of the number of muLECs endocytosed dye as a percentage of total muLECs (**m**) and the length of muLEC-derived vessels draining fluid as a percentage of total muLEC-derived vessels (**n**) after ICV or DA injection of Alexa647-dextran. 12 fish were observed in three independent experiments in each group. Error bars, mean ± SEM. Unpaired two-tailed Student's *t*-test. *P* values included in the graphs. Source data are provided as a Source Data file. The white dashed boxes outline the enlarged areas. Scale bars: 50 μm.

capacity of endocytosis, but instead perform liquid transport function in a tubular form (Fig. 6f–h). Thus, mLSCs maintain the dispersed morphologies of muLECs under physiological conditions.

The membrane-like structures formed by mLSCs (Fig. 4b) are likely to be fundamental for them to support the development, maintenance, and regeneration of muLECs. The migration and growth of one cellular structure could be dependent on the guidance and support provided by another pre-formed cellular structure[17,43,44]. This mechanism could explain how the pre-formed framework of mLSCs provides support and guidance for the sprouting and post-injured regeneration of muLECs. The mLSCs express several pro-lymphangiogenic factors and their ablation severely impairs muLEC

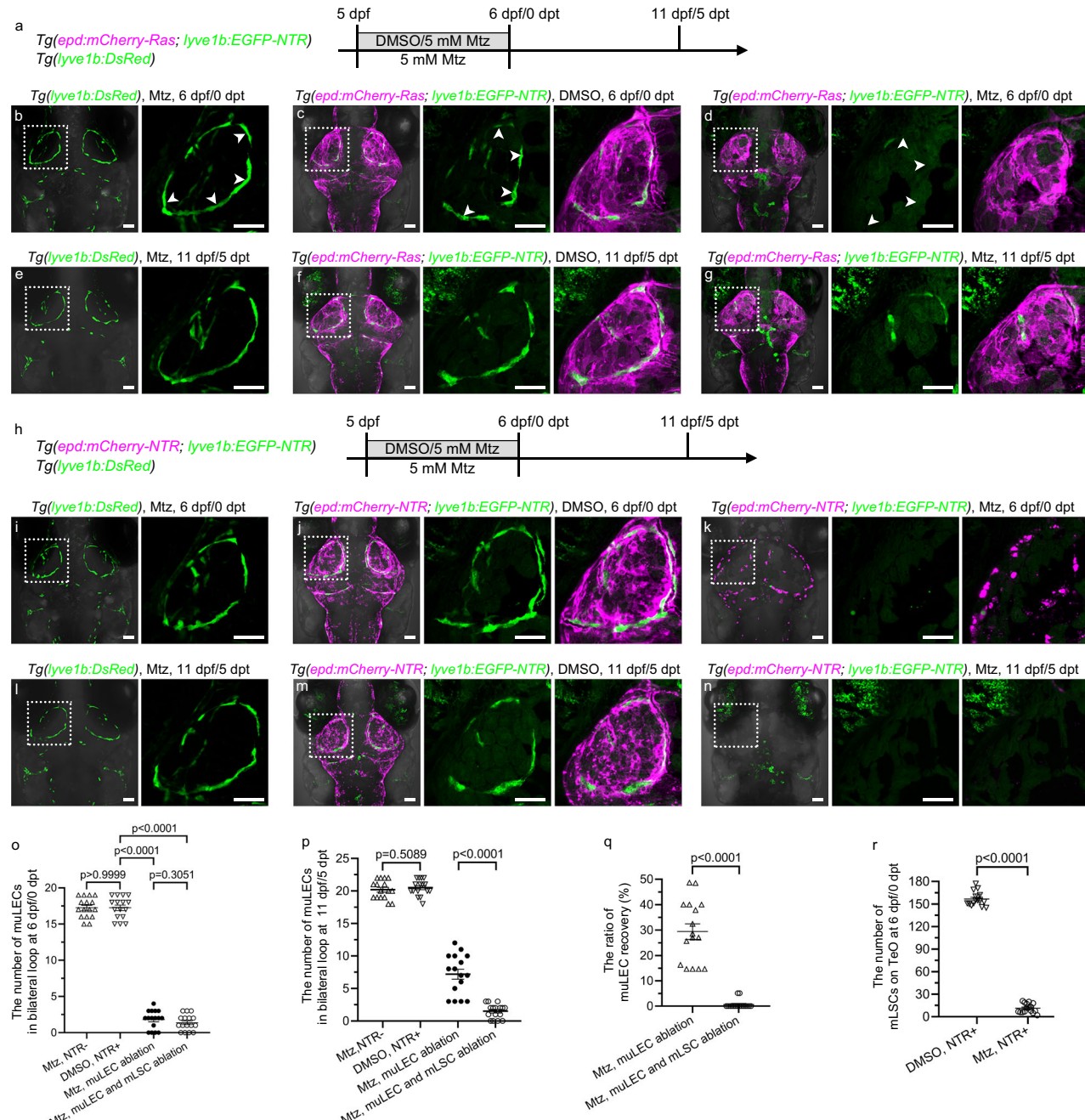

**Fig. 7 | The mLSCs are necessary for the regeneration of muLECs after injury.**
**a, h** Experimental design for ablation of meningeal lymphatic supporting cells (mLSCs) and mural lymphatic endothelial cells (muLECs). **b, c, e, f, i, j, l, m** Dorsal confocal images of muLECs and mLSCs in unablated brains at 6 dpf/0 dpt and 11 dpf/5 dpt. *n* = 37 per experiment. White arrowheads indicate muLECs. **d** Dorsal confocal images of muLEC ablation at 6 dpf/0 dpt. White arrowheads indicate muLECs after ablation. *n* = 37. The experiment was repeated three times independently with similar results. **g** Dorsal confocal image of partial regeneration of muLECs after injury at 11 dpf/5 dpt. *n* = 28. The experiment was repeated three times independently with similar results. **k** Dorsal confocal images of muLEC and mLSC ablation at 6 dpf/0 dpt. *n* = 39. The experiment was repeated three times independently with similar results. **n** Dorsal confocal images of less regeneration of muLECs after muLEC and mLSC ablation at 11 dpf/5 dpt. *n* = 34. The experiment was

repeated three times independently with similar results. **o, p** Quantification of the number of muLECs in bilateral loop in the nonablation, muLEC ablation, and muLEC and mLSC double ablation larvae at 6 dpf/0 dpt (**o**) and at 11 dpf/5 dpt (**p**). 16 fish were observed in three independent experiments in each group. **q** Quantification of the recovery rate of muLECs in bilateral loop after muLEC ablation and double muLEC and mLSC ablation. 16 fish were observed in three independent experiments in each group. The same larvae as in (**o, p**) were used. **r** Quantification of the number of mLSCs on the optic tectum (TeO) for larvae with unablated and ablated mLSCs at 6 dpf/0 dpt. 16 fish were observed in three independent experiments in each group. Error bars, mean ± SEM. Unpaired two-tailed Student's *t*-test. *P* values included in the graphs. Source data are provided as a Source Data file. The white dashed boxes outline the enlarged areas. Scale bar: 50 μm.

development. However, at later stages, many other cell types including BECs, smooth muscle cells, macrophages, and fibroblasts could also produce pro-lymphangiogenic factors[45,46]. Even muLECs themselves could express some pro-lymphangiogenic factors such

as Vegfc, Vegfd, and Mmp2[15]. After ablation of mLSCs at 5 dpf, the formation of lymphatic vessels by muLEC interconnection and lumenization (Fig. 5) is a relatively slow process compared to the outgrowth of muLECs, and this process may be assisted by non-

mLSC-derived pro-lymphangiogenic factors. The process of muLEC interconnection and lumenization to form lymphatic vessels may involve more complex mechanisms, which need further investigation. Furthermore, after the ablation of mLSCs, the muLECs that lose support from mLSCs gradually transform into functional lymphatic vessels rather than undergo cell death. These observations indicate that the morphologies but not survival of muLECs are dependent on mLSCs, also suggesting the muLEC plasticity. These findings provide compelling evidence that muLECs represent a distinct LEC lineage and help to clarify the plasticity, heterogeneity, and special functions of LECs in the meninges.

Previous scRNAseq data of whole embryos/larvae and adult telencephalon showed that *epd* is specifically expressed in one-cell cluster[34–36], which also expresses other mLSC markers used in our study. Here, we validated the transcriptomic specificity of mLSCs by re-analyzing scRNAseq data from Farnsworth et al.[34]. Besides, we verified that mLSCs in embryos/larvae did not express common meningeal cell markers by bulk RNA sequencing (Fig. 3e) and confirmed morphological differences between mLSCs and other common meningeal cells using multiple transgenic lines (Fig. 2). Furthermore, apoptosis assays confirmed that ablation of mLSCs do not lead to widespread cell death, especially not in BECs and muLECs (Supplementary Figs. 7 and 9). Thus, these findings provide evidence for the transcriptional and anatomical specificities of mLSCs.

Although Ependymin is a protein specific in fish, several molecular markers of mLSCs are conserved in mammals such as *slc13a4*, *igfbp2a*, and *nid1b*, providing preliminary evidence to implicate the existence of a similar cell type in mammals. It is important to note that the mammalian leptomeninges feature a more complex structure and cellular composition than those of zebrafish, therefore the mLSC marker-expressing cells in mammals could format a different structure and carry out functions in a more complex manner. After the discovery of meningeal lymphatics, recent findings of novel lymphatic functions in several tissues and identification of related cell types further enhance our understanding of lymphatic diversities and functional heterogeneities in different organs and tissues[47–50]. For example, recent findings that lymphatics in bone are important for bone regeneration have added new insights into the organ-specific functions and molecular specialization of lymphatics[47].

## Methods

### Study approval
All animal experiments were approved by the Institutional Animal Care and Use Committee (IACUC) of Southwest University Laboratory Animal Center. All animal procedures followed standard conditions in accordance with the regulations of the Ethics Committee of Southwest University (Chongqing, China).

### Zebrafish handling and strains
All zebrafish lines were maintained and raised under standard laboratory conditions. In order to inhibit pigmentation, zebrafish embryos were treated with 0.003% 1-phenyl-2-thiourea (PTU, Sigma−Aldrich) from 24 hpf, or alternatively, their parents were crossed into the *casper* background.

The transgenic zebrafish lines generated in this study were *Tg(epd:EGFP)*[cq188], *Tg(epd:EGFP-NTR)*[cq189], *Tg(epd:mCherry-Ras)*[cq190], *Tg(epd:mCherry-NTR)*[cq191], *Tg(epd:H2B-GFP)*[cq192], *Tg(epd:H2B-mCherry)*[cq193], *Tg(lyve1b:EGFP-NTR)*[cq194], and *Tg(sox10:EGFP-NTR)*[cq195]. The previously published zebrafish lines used were *Tg(lyve1b:DsRed)*[cq27] (refs. 17,21), *Tg(lyve1b:EGFP)*[cq86] (ref. 18), *Tg(abcc9*[BAC]*:Gal4ff)*[ncu34] (ref. 25), *Tg(pdgfrb*[BAC]*:GFP)*[ncu22] (ref. 26), *Tg(acta2:GFP)*[ca7] (ref. 27), *Tg(coro1a:Kaede)*[cq22] (ref. 28), *Tg(lyz:GFP)*[nz117] (ref. 29), *Tg(mpeg1:GFP)*[gl22] (ref. 30), *Tg(prox1a*[BAC]*:KalTA4; UAS:TagRFP)*[nim5] (ref. 31), *Tg(nkx2.2a:GFP)*[ia3] (ref. 32), *Tg(elavl3:GFP)*[knu3] (ref. 33), *Tg(kdrl:mCherry-Ras)*[s896] (ref. 51), *Tg(fli1:GFP)*[y1] (ref. 52), *Tg(kdrl:GFP)*[s843] (ref. 53), *Tg(fli1:nEGFP)*[y7] (ref. 54), *and casper*[55]. All the

lines used in this study were stable and breeding transgenic lines. The strains, numbers, and ages of zebrafish used in each experiment are described in the corresponding figures and legends. All zebrafish were used with an equal number of males and females when sex determination has occurred.

### Generation of plasmids and transgenic lines
To generate reporter lines driven by the *epd* promoter, the ~3.9 kb upstream DNA sequence of the *epd* gene was amplified from the wild-type zebrafish genomic DNA. The ApaI and AgeI digest sites were added to the 5′ and 3′ end of the amplified sequence, respectively. The *epd* promoter DNA was digested with ApaI and AgeI enzymes, serving as the insert fragment. Additionally, the plasmid pBluescript2KS(-)_krt18:EGFP-NTR was used as the vector, which had been digested with ApaI, AgeI, and EcoRI enzymes. Subsequently, the digested fragments were ligated with T4 DNA ligase resulting in the formation of the plasmid pBluescript2KS(-)_epd:EGFP-NTR. The sequences encoding EGFP, mCherry-NTR, H2B-mCherry, H2B-GFP, and mCherry-Ras were amplified by PCR before being inserted into the 3′ end of *epd* promoter through subcloning, resulting in the generation of *epd:EGFP*, *epd:mCherry-NTR*, *epd:H2B-mCherry*, *epd:H2B-GFP*, and *epd:mCherry-Ras* constructs. These plasmids were respectively co-injected with *I-SceI* and 10× *I-SceI* buffer into one-cell-stage wild-type zebrafish embryos to obtain F0. F1 was screened from the descendant embryos generated from the cross of F0 and wild-type based on the correct expression position of the respective fluorescent proteins.

For the generation of *Tg(lyve1b:EGFP-NTR)* zebrafish line, the plasmid pT2KXIGDin_lyve1b:EGFP was digested with ClaI enzyme and the plasmid pBluescript2KS(-)_cldn15lb-1:EGFP-NTR was digested with NotI-HF enzyme. Then, the two digested DNA fragments were blunted with Klenow to eliminate the sticky ends created by the endonucleases. Subsequently, both blunted DNA fragments were subjected to AgeI-HF digestion. Finally, the produced vector fragment pT2KXIGDin_lyve1b and the insert fragment EGFP-NTR were ligated with T4 DNA ligase to obtain the plasmid pT2KXIGDin_lyve1b:EGFP-NTR. F0 was generated by co-injecting the plasmid pT2KXIGDin_lyve1b:EGFP-NTR with *Tol2* transposase RNA (40–50 pg) into one-cell-stage wild-type zebrafish. F0 and wild-type zebrafish were crossed to select F1 possessing the correct fluorescence expression.

To generate the *Tg(sox10:EGFP-NTR)* zebrafish line, the ~5.1 kb *sox10* promoter was amplified from the genomic DNA of wild-type zebrafish. The insert fragment was obtained by digesting the *sox10* promoter with ApaI and AgeI enzymes, while the vector fragment was obtained by digesting the plasmid pBluescript2KS(-)_krt18:EGFP-NTR with ApaI, AgeI, and EcoRI enzymes. Subsequently, T4 DNA ligase was used to ligate the insert and vector fragments, resulting in the generation of the plasmid pBluescript2KS(-)_sox10:EGFP-NTR. F0 was generated by co-injecting the plasmid pBluescript2KS(-)_sox10:EGFP-NTR with *I-SceI* and 10× *I-SceI* buffer into wild-type zebrafish embryos at one-cell stage. F1 with correct fluorescence expression was selected from the offspring of the cross between F0 and wild-type zebrafish.

Information about primers for plasmid construction is provided in Supplementary Table 1.

### Whole-mount in situ hybridization (WISH) and antibody staining
WISHs were performed as previously described[56]. Briefly, larval zebrafish were fixed in 4% paraformaldehyde (PFA) at 4 °C for 24 h. They were then dehydrated with 100% methanol and incubated at −30° overnight. The dehydrated larvae were rehydrated with methanol and 1× PBT (1× PBS with 0.1% Tween 20). Rehydrated larvae were then digested with proteinase K (10 μg/ml in 1× PBT) at room temperature (RT) for 30 min. Followed by incubation with 4% PFA again for half an hour at RT. Then digested larvae were prehybridized

with 100% HYB at 68.5 °C for 5 h and hybridized with HYB-containing probe at 68.5 °C overnight. Subsequently, the larvae were washed at 68.5 °C with preheated HYB, 2× SSCT, and 0.2× SSCT and washed with 0.2× SSCT and 1× MABT at RT. Then they were blocked with blocking buffer for 2 h at RT and incubated with Anti-digoxigenin AP, Fab fragment (1:2000, 11093274910, Roche) at 4 °C overnight. Finally, the larvae were washed with 1× MABT and NTMT at RT and incubated with NBT/BCIP solution (11681451001, Roche) (0.2% NBT/ BCIP stock solution in 1× MABT) at 37 °C. For probes (*epd*, *ggctb*, *slc13a4*, *apof*, *nid1b*, *slc7a2*, *soul5*, and *igfbp2a*), templates were acquired by PCR amplification of cDNA with a T7 promoter sequence from 3-dpf AB strains. Information about primers for probe synthesis is provided in Supplementary Table 1.

Antibody staining was performed following previously established protocols[57–59]. Briefly, zebrafish larvae and dissociated adult brains were fixed in 4% formaldehyde at 4 °C for 24 h. Larval zebrafish were dissociated to obtain brains after being fixed. Subsequently, the specimens were rinsed in 1× PT (1× PBS with 1% TritonX-100) at RT and blocked with 1× PBTN (1× PT with 4% BSA) at 4 °C for 2 h. Then they were incubated with primary antibodies at 4 °C overnight. Primary antibodies used were: anti-GFP (1:2000, ab6658, Abcam), anti-mCherry (1:2000, ab125096, Abcam), anti-DsRed2 (1:2000, sc-101526, Santa Cruz), and Anti-Collagen I (1:1000, ab23730, Abcam). After that, the specimens were washed with 1× PT at RT and then stained with secondary antibodies at 4 °C overnight. Secondary antibodies used were: Donkey anti-goat IgG Alexa fluor 488-conjugated (1:2000, A11055, Invitrogen), Donkey anti-mouse IgG Alexa fluor 568-conjugated (1:2000, A10037, Invitrogen), and Donkey anti-rabbit IgG Alexa fluor 568-conjugated (1:2000, A10042, Invitrogen). Finally, the specimens were washed with 1× PT and then stained with DAPI (D8417, Sigma–Aldrich) at RT for 30 min. Information for the antibodies is summarized in Supplementary Table 2.

### Fluorescence in situ hybridization (FISH) combined with antibody staining (FISH-antibody staining)
FISH-antibody stainings were performed as previously described[60]. In brief, after being fixed in 4% PFA for 24 h, the zebrafish larvae were dissected and their brains were obtained and dehydrated with 100% methanol. Subsequently, the brains were rehydrated with methanol and 1× PBT, then digested with proteinase K at RT for 5 min, followed by incubation with 4% PFA again at RT for 20 min. After that, the brains were prehybridized with 100% HYB at 65 °C for 5 h, and then hybridized with a HYB-containing probe at 65 °C overnight. Then, the brains were washed with preheated HYB, 2× SSCT, and 0.2× SSCT at 65 °C, and washed with 0.2× SSCT and 1× MABT at RT. Followed by blocked brains with blocking buffer and incubated with Anti-digoxigenin POD, Fab fragment (1:2000, 11207733910, Roche) at 4 °C overnight. The brains were then washed with 1× MABT and 1× PBS at RT and then incubated with Cyanine 3 in amplification diluent buffer (1:50, NEL701A001KT, PerkinElmer) at RT overnight. The subsequent procedures were performed according to the antibody staining procedure. The probes used were identical to those employed in WISHs. The antibodies utilized were anti-GFP (1:2000, ab6658, Abcam) and Donkey anti-goat IgG Alexa fluor 488-conjugated (1:2000, A11055, Invitrogen).

### Vibratome section of zebrafish brains
The fixed adult brains of zebrafish were mounted in the 4% low melting agarose and sliced at 200 μm using LEICA VT1000S vibratome.

### Tissue isolation, FAC sorting, and data analysis for RNA sequencing of mLSCs and zebrafish embryos/larvae
Heads of transgenic *Tg(epd:EGFP-NTR; lyve1b:DsRed)* zebrafish at 55 hpf and 5 dpf were separately dissected and placed in 1 mL 1× PBS on ice. They were then washed with 1 mL 1× PBS once and centrifuged at

4000 × *g* at 4 °C for 2 min. Next, the supernatant was removed and heads were dissociated with a mixture solution of 200 μL PBS-EDTA (1 mM EDTA in 1× PBS) and 50 μL 2.5% trypsin. The homogenized cell suspension was centrifuged at 4000 × *g* at 4 °C for 2 min before being washed with 1 mL 1× PBS twice. The cells were then resuspended in 200 μL 1× PBS within 5 min and the cell suspension was collected by filtering through a 40 μm cell strainer into a 2 mL EP tube. Wild-type zebrafish heads at 55 hpf and 5 dpf were operated using the same dissociation procedure as a fluorescent-negative control for FAC sorting. Subsequently, cell sorting was performed using flow cytometry (Moflo XDP, Beckman) to obtain 55 hpf and 5 dpf of EGFP-NTR (+) DsRed (−) cells as two biological replicates. Next, single-cell transcriptomic amplification was performed on the obtained cells. All tissue isolation procedures were carried out on ice. The sorted cells were harvested directly in a single-cell collection solution containing cell lysis components and RNase inhibitors. The 1st cDNA was generated via reverse transcription using a nucleic acid sequence with Oligo dT, followed by PCR amplification to enrich the nucleic acid and purify the amplified product for library construction. For control tissues of 55 hpf or 5 dpf, there were three biological replicates per period, each containing one hundred zebrafish. Total RNA was extracted from whole zebrafish by Trizol-Chloroform-Isopropanol isolation and then purified by magnetic beads with Oligo dT. The purified mRNA was fragmented by adding a fragmentation buffer to create short fragments, which were then used as templates for constructing cDNA libraries. Constructed libraries were sequenced using the Illumina platform with the sequencing strategy PE150. The raw data aligned to the zebrafish (Danio rerio) reference genome (GRCz11.96). Data analysis was carried out as previously described[28]. In brief, Clean Reads are obtained by processes such as removing Raw Reads from low-quality sequences to complete data processing, and all subsequent analysis is based on Clean Reads. Gene enrichment ontology analysis of highly expressed genes in *epd*-positive cells relative to whole fish was performed using Metascape (https://metascape.org/gp/index.html#/main/step1)[61]. The top 500 protein-coding genes in 55 hpf as well as 5-dpf *epd*-positive cells with padj <0.05 and ranked in descending order by Log2Fold-Change were selected for gene enrichment ontology analysis.

### Data analysis of single-cell RNA sequencing
The single-cell RNA-sequencing (scRNAseq) data analyzed in this paper were mined from Farnsworth et al.[34]. ScRNAseq data were preprocessed and normalized using the R package "Seurat v4.3.0". The *epd*-positive cells, fibroblasts, mural cells, neural progenitor cells, immune cells, and endothelial cells were picked out for further analysis according to the classical marker genes of each cell type. The Uniform Manifold Approximation and Projection (UMAP) plot was obtained by dimensionality reduction analysis using the RunU-MAP function (R package "Seurat v4.3.0"). The FindAllMarkers function (R package "Seurat" v4.3.0) was used to identify the highly expressed genes in each cluster, and the genes with padj <0.05 were taken to be sorted by avg_log2FC, and the genes in the top 50 avg_log2FC were taken to be plotted by heatmap (R package "pheatmap 1.0.12"), take the genes in the top 20 of avg_log2FC of *epd*-positive cell population and plot them by dotplot (R package "Seurat v4.3.0"). Violin plot of each cell-specific marker gene using the VlnPlot function in the R package "Seurat v4.3.0". For Principle component analysis (PCA), the *epd*-positive cells in scRNAseq data were first split into 3 groups for pseudo-bulk process according to the mean value of expression, then ComBat_seq in R package "sva 3.42.0" was used to eliminate the batch effect on bulk RNA-sequencing data (*epd*-positive cells at 55 hpf and 5 dpf, and whole zebrafish at 55 hpf and 5 dpf) generated in this study and the *epd*-positive cells in single-cell RNA-sequencing data that were split into 3 groups, and finally PCA clustering analysis was performed with R package "ape 5.7.1".

## Imaging

Larvae in which whole-mount in situ hybridization was performed were imaged using a ZEISS SteREO Discovery.V20 microscope. Antibody-stained brains or brain sections, FISH-antibody-stained brains, live embryos, live larvae, or live juveniles were mounted in 1% low melting point agarose. They were then imaged with either a ×10 air objective or a ×20 water immersion objective installed in the ZEISS LSM780 or LSM 880 confocal microscope. Live embryos or larvae were mounted in 1% low melting point agarose and then time-lapse imaged using the Zeiss Lightsheet Z.1 or ZEISS LSM 880 confocal microscope.

## Chemical treatment

In the treatment of metronidazole (Mtz, Sigma–Aldrich), the Mtz was completely dissolved in egg water that contained 0.2% dimethyl sulfoxide (DMSO). Within a *Tg(epd:EGFP-NTR)* or *Tg(epd:mCherry-NTR)* background, embryos at 48 hpf and larvae at 5 dpf were incubated with 10 mM Mtz for 24 h, while juveniles at 25 dpf were incubated with 5 mM Mtz for 22 h; within a single *Tg(lyve1b:DsRed)* background, embryos at 48 hpf were incubated with 10 mM Mtz for 24 h, while larvae at 5 dpf were incubated with 5 mM Mtz or 10 mM Mtz for 24 h; within a *Tg(sox10:EGFP-NTR)* background, larvae at 5 dpf were incubated with 5 mM Mtz for 24 h; within a *Tg(lyve1b:EGFP-NTR)* background, larvae at 5 dpf were incubated with 5 mM Mtz for 20 h. For muLEC ablation, larvae with a large number of muLECs ablated on TeO and some other muLECs retained need to be selected for analysis. Each ablated group was incubated with 0.2% DMSO in the egg water as a control. Subsequently, the embryos, larvae, or juveniles were washed three times with egg water and recovered in egg water, and marked as 0 dpt.

## Dye injection

The dye injection was carried out as previously described[14,17,18]. Briefly, the dye is injected directly into the dorsal aorta or into the intracerebroventricular of zebrafish using a glass capillary needle. The dyes used were IgG-conjugated Alexa fluor 647 (150 kDa) (2 mg/ml, A31573, Invitrogen) and Alexa647-Dextran (10 kDa) (2 mg/ml, D22914, Invitrogen).

## TUNEL assay

The TUNEL assay for the detection of cell apoptosis was performed as previously described[17]. In brief, zebrafish larvae underwent fixation, dehydration, rehydration, brain dissected, and digestion in accordance with the procedures of FISHs. Then the brains were treated with acetone at −20 °C for 1 h and subsequently were fixed with 4% PFA for 20 min at RT. The brains were sequentially incubated with buffer and enzyme from In Situ Cell Death Detection Kit (12156792910, Roche) at 37 °C for 2 h and then were washed with preheated 2× SSCT at 37 °C and washed with 2× SSCT and 1× PBT at RT.

## Statistics and reproducibility

All confocal imaged pictures and WISHs represented at least 3 independent experiments with similar results. The numbers of animals represented by the images were indicated in the corresponding figure legends. The number of specimens used for RNA sequencing along with the number of biological parallels were described in the corresponding figure legends. The statistical calculations were performed using GraphPad Prism 8.0.1. The statistical data were analyzed by unpaired two-tailed Student's *t*-test and two-way ANOVA Sidak's multiple comparisons test. All "*n*" and "*P*" values as well as statistical tests are presented in the corresponding figures and legends. The results with *P* values less than 0.05 were considered to be statistically significant. With the exception of Fig. 7q, Supplementary Fig. 8a–d, each dot and number in the remaining statistics figures represents an independent fish. All movies were handled with Fiji (ImageJ) 2.14.0 and ZEN2010 Imaging software. The heatmaps in Fig. 3e, f were created using Heatmap Illustrator (HemI) version 1.0.3.7 software[62].

## Reporting summary

Further information on research design is available in the Nature Portfolio Reporting Summary linked to this article.

## Data availability

The RNA-sequencing data generated in this paper have been deposited in the Genome Sequence Archive in the National Genomics Data Center under the accession code GSA: CRA016778. Previous published scRNAseq data that were re-analyzed in this study are available in the National Centre for Biotechnology (NCBI) SRA: PRJNA564810[34]. All other relevant data that support the findings of this study are available within the article, its Supplementary Information, and its Supplementary Data, or from the corresponding author upon reasonable request. Source data are provided with this paper.

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

## Acknowledgements

We thank J He, J Chen, Y Yang, X Wei, P Cai for helpful discussions. This work was supported by the National Key R&D Program of China (2021YFA0805000) to L.L., and the National Natural Science Foundation of China (32192400) to L.L.

## Author contributions

L.L., X.H., and D.X. designed the experimental strategies, analyzed data, and wrote the manuscript. L.Z. conducted analyses of RNA-sequencing data. J.F. generated the pericyte-labeling transgenic line. X.H. and D.X. performed all the other experiments in this study.

## Competing interests

The authors declare no competing interests.
