## [Peer Review File · Nature Communications]

Meningeal lymphatic supporting cells govern the formation and maintenance of zebrafish mural lymphatic endothelial cellsEditorial Note: Parts of this Peer Review File have been redacted as indicated to remove third-party material where no permission to publish could be obtained.

REVIEWER COMMENTS

Reviewer #1 (Remarks to the Author):

In the manuscript by He and colleagues entitled "Meningeal lymphatic supporting cells govern the formation, maintenance and regeneration of zebrafish mural lymphatic endothelial cells", the authors describe a new, ependymin expressing, supporting cells in the zebrafish leptomeninges that support the development and maintenance of the muLECs. Even more interestingly, the author show that the mLSCs are responsible for the lack of lumenization of the muLECs. While the study is very interesting, thorough in its description and visually stunning, the current version of the manuscript could benefit some improvements prior to publications:

Major points:

- The major drawback from this manuscript is the lack of quantifications. While this reviewer appreciate the representative images and video being provided, quantifications would greatly enhance the message the authors are trying to deliver. In Fig 4, quantification of the efficiency of ablation, and number of muLECs would be required to validate the reproducibility and the variability of the experiment. The same applied for Fig 5; 6 and 7. In Supplementary figure 5; when the authors analyze the effects of the ablation on the local blood endothelium, the authors should similarly provide quantifications. Particularly since previous work demonstrated that lack of muLECs results in vascular impairment. If the author's data is somewhat contradicting previous publication, quantification of parameters of the blood vasculature would be required.
- The functional characterization of the newly form muLECs lymphatic vessels in Fig 6 is not the most convincing. The choice of color, particularly, makes it extremely difficult to see the blue tracer on the images. Quantifications and more legible representative images would greatly re-inforce the message.
- In Figure 2, the authors are trying to identify the lineage the edp+ cells belong to. They rule out every cell type they tested, including fibroblasts. The staining with collagen is only performed at 6mpf, a different time point from every other markers they have looked at; do the authors anticipate a different staining at earlier time point ? Since Collagen I is a staining on a cross section, is it possible that the staining is of the deposited collagen rather than of the cell that expresses it? If not fibroblast/stromal cells, what other meningeal cell type do the author think these cells are ? There seems to be a missed opportunity from using the publicly available single cell RNA seq that the authors are referencing. Re-analysis of that dataset to visualize the edp+ cells and the potential markers it would express would greatly benefit the authors in their quest to identify the lineage these cells belong to. Reanalysis of that data would also corroborate some of the analysis present in that paper (Fig 3).
- There is some confusing statement in regards to Fig 3. The text implies that LECs were also isolated for RNA sequencing, but the data is not presented anywhere in the figure? is that correct? The gating strategy to identify the GFP and DsRed positive cells is also confusing and appear to be cutting through the actual positive cell populations. How did the authors decide on their gating. Were non transgenic fish used to validate the gating ? A pathway analysis combining genes identified in this experiment and the ones from the published RNA sequencing would also provide the authors with a stronger idea of how the mLSCs regulate muLECs. Indeed the mLSCs seem to be a major source of lymphangiogenic markers, yet their depletion results in formation of lymphatic vessels. These results suggest that the mLSCs are not a soul source of lymphangiogenic factors but rather serve a much greater role that is not explained via the presented sequencing experiments.

Minor point:

- In the introduction, the authors are referencing a manuscript claiming that muLECS are also present

in mammals. This reviewer would caution the authors to be more hypothetical when translating the presence of muLECs to mammals as that paper is controversial and the presence of true muLECs in mammals remains extremely debatable.

- This reviewer sees it as a missed opportunity to not hypothesize about the functional role of preventing muLECs to form tubular structure during normal development in the discussion section.

Reviewer #2 (Remarks to the Author):

Recent studies have shed light on the heterogeneous compositions of lymphatic endothelial cells (LECs) and their unique functions in specific tissues. One such population is the mural lymphatic endothelial cells (muLECs), which do not form vessel structures but remain dispersed, facilitating their clearance of metabolic waste from the brain. Despite the crucial functions of muLECs in brain immune homeostasis, how they form during development and how they maintain the unique morphology remain largely unknown. In this manuscript, He et al. identified a distinct meningeal cell population which regulates the development, morphology maintenance, and regeneration of muLECs.

To begin with, the authors generated a specific reporter line for this cell population based on its specific expression of the gene *epd*. Using this reporter line, they comprehensively characterized the cellular and molecular features of this unique cell population. They demonstrated that these cells represent a previously unidentified population distinct from other mural cell types. They further discovered that the *epd*⁺ cells are closely associated with blood vessels and muLECs during development and possess a pro-lymphangiogenic transcription profile, suggesting their potential role in muLEC formation and function. To explore the function of the *epd*⁺ cells, the authors specifically ablated these cells at different developmental stages and conditions and examined their impact on muLECs. Remarkably, they found that the loss of the *epd*⁺ cells led to the blockade of muLEC formation during development and disrupted their morphology and regenerate at later stages. These findings demonstrate an essential role of the *epd*⁺ cells in supporting muLEC development and function. Hence, the authors named these cells as meningeal lymphatic supporting cells (mLSCs). In sum, the authors conducted elegant studies on this novel and important cell population. The experiments were well-designed, and the results were clearly presented. The major conclusions are well-supported by the experimental data. Overall, it is a great piece of work and well-suited for publication in Nature Communications. I have only a few minor comments and some suggestions.

1, The *epd*⁺ cells appear to be ubiquitously distributed and form a membranous structure that covers the entire meninges. Yet a substantial portion of the *epd*⁺ cells do not seem to directly connect to muLECs, raising the possibility that these cells may have additional functions beyond supporting muLECs. It would be valuable if the authors could discuss this issue in the discussion.

2, In the cell depletion assays, it seems that the *epd*⁺ cells could not be recovered even after the removal of Mtz. Does this suggest that these cells are maintained and expanded by self-proliferation during development? It would be valuable for the authors to discuss the potential origin and lineage development of this cell population.

3, In Fig. 3b, the sorting strategy for the target *epd*⁺ cells is not clearly depicted, and there is a lack of a fluorescence-negative control for gating.

4, Line 51 and 66: the full name of muLECs and mLSCs should be provided when it is first mentioned in the main text.

Point-by-Point Response to the Reviewers

Reviewer #1 (Remarks to the Author):

In the manuscript by He and colleagues entitled "Meningeal lymphatic supporting cells govern the formation, maintenance and regeneration of zebrafish mural lymphatic endothelial cells", the authors describe a new, ependymin expressing, supporting cells in the zebrafish leptomeninges that support the development and maintenance of the muLECs. Even more interestingly, the author show that the mLSCs are responsible for the lack of lumenization of the muLECs. While the study is very interesting, thorough in its description and visually stunning, the current version of the manuscript could benefit some improvements prior to publications:

Major points:

1. The major drawback from this manuscript is the lack of quantifications. While this reviewer appreciate the representative images and video being provided, quantifications would greatly enhance the message the authors are trying to deliver. In Fig 4, quantification of the efficiency of ablation, and number of muLECs would be required to validate the reproducibility and the variability of the experiment. The same applied for Fig 5; 6 and 7. In Supplementary figure 5; when the authors analyze the effects of the ablation on the local blood endothelium, the authors should similarly provide quantifications. Particularly since previous work demonstrated that lack of muLECs results in vascular impairment. If the author's data is somewhat contradicting previous publication, quantification of parameters of the blood vasculature would be required.

Response : We have followed the reviewer's suggestion to provide quantifications in the revised manuscript and updated the corresponding statements according to the quantifications (Fig. 4-7; page 13-15, 18, line 254-256, 280-287, 352-353). In Supplementary Fig. 9 (previously Supplementary Fig. 5), we have included statistics for

diameter and cell number of mesencephalic vein (MsV). Ablation of mLSCs did not cause impairment of brain vascular maintenance (Supplementary Fig. 9a-f; page 16, line 310-315). However, we have not studied the effects of mLSCs on early brain vascular development in this manuscript. Thus, we have NOT presented any data contradicting previous publications. Additionally, ablation of mLSCs resulted in the change of muLEC morphologies, but did not cause muLEC apoptosis and reduction in numbers (Supplementary Fig. 7b-d).

2. The functional characterization of the newly form muLECs lymphatic vessels in Fig 6 is not the most convincing. The choice of color, particularly, makes it extremely difficult to see the blue tracer on the images. Quantifications and more legible representative images would greatly reinforce the message.

Response: We have replaced the images in the revised Figure 6 with more legible representative images and have adjusted the color of the tracer from blue to a more visible white color. Meanwhile, we have also included statistics for Figure 6.

3. In Figure 2, the authors are trying to identify the lineage the edp+ cells belong to. They rule out every cell type they tested, including fibroblasts. The staining with collagen is only performed at 6mpf, a different time point from every other markers they have looked at; do the authors anticipate a different staining at earlier time point ? Since Collagen I is a staining on a cross section, is it possible that the staining is of the deposited collagen rather than of the cell that expresses it? If not fibroblast/stromal cells, what other meningeal cell type do the author think these cells are ? There seems to be a missed opportunity from using the publicly available single cell RNA seq that the authors are referencing. Re-analysis of that dataset to visualize the edp+ cells and the potential markers it would express would greatly benefit the authors in their quest to identify the lineage these cells belong to. Reanalysis of that data would also corroborate some of the analysis present in that paper

Response: In the Response Figure 1 below, we have shown anti-Collagen I antibody staining at 8 dpf and *in situ* hybridization using

col1a1a probe at 5 dpf. The Collagen I+ fibroblast was undetectable in the larval brain at these stages (Response Fig 1. a, b). That is why we presented data of adult zebrafish at 6 mpf. Cross-section images were provided in order to ensure whether mLSCs and the Collagen I+ fibroblasts were co-stained (Fig. 2c). Since they stained different layers and both layers were co-stained with DAPI, we have drawn the conclusion that mLSCs and the Collagen I+ fibroblasts represent different but adjacent layers. We have also provided whole mount staining images of mLSCs and the Collagen I+ fibroblasts below, but these images are not informative (Response Fig. 1).

Finally, according to the reviewer's suggestion, we have re-analysed publicly available scRNAseq data from Farnsworth et al.¹. Please see Supplementary Fig. 3 and page 9-10, line 171-184, 192-194 of the revised manuscript for detailed information. Single-cell sequencing data from Sur et al. at multiple time points between 3 hpf-120 hpf in whole zebrafish embryos classified mLSCs into a subclass of mesenchyme distinct from fibroblasts². We have included here a Uniform manifold approximation and projection (UMAP) diagram of the mesenchyme they defined (Response Fig. 2; Images modified from <https://daniocell.nichd.nih.gov/>).

Response Figure 1: The *epd*-positive cells do not correspond to classical fibroblasts. **a** WISH of *col1a1a* in wild-type larvae at 5 dpf. The lateral view of the frame region is shown. $n = 15/15$. Scale bars: 50 μm . **b** Lateral confocal images of larval brain show expression of fibroblast marker collagen I in the brain at 8 dpf. The boxed areas are magnified. $n = 12/12$. Scale bars: 50 μm . **c** Lateral confocal image of the larval trunk show Collagen I labelled fibroblasts. $n = 12/12$. Scale bars: 50 μm . **d** Dorsal confocal images

of the adult brain show *epd*-positive cells and collagen I-labelled fibroblasts at 6 mpf. $n = 2$ male and 2 female adults. The enlarged boxed area showed *epd*-positive cells and fibroblasts are not co-localized. Scale bars: 50 μm

[redacted]

Response Figure 2. a, b, c UMAP plots showing the mLSCs (a), fibroblast 1 (b), and fibroblast 2 (c) in the mesenchyme (<https://daniocell.nichd.nih.gov/>).

4. There is some confusing statement in regards to Fig 3. The text implies that LECs were also isolated for RNA sequencing, but the data is not presented anywhere in the figure? is that correct? The gating strategy to identify the GFP and DsRed positive cells is also confusing and appear to be cutting through the actual positive cell populations. How did the authors decide on their gating. Were non transgenic fish used to validate the gating ? A pathway analysis combining genes identified in this experiment and the ones from the published RNA sequencing would also provide the authors with a stronger idea of how the mLSCs regulate muLECs. Indeed the mLSCs seem to be a major source of lymphangiogenic markers, yet their depletion results in formation of lymphatic vessels. These results suggest that the mLSCs are not a soul source of lymphangiogenic factors but rather serve a much greater role that is not explained via the presented sequencing experiments.

Response: The gating strategy to identify the *epd*:GFP+ and *lyve1b*:DsRed+ cells was designed to accurately separate mLSCs from closely-contacted muLECs. Although the *lyve1b*:DsRed+ were isolated in FACS, they were NOT collected and subjected to RNA sequencing. We have corrected the description of RNA sequencing in the revised manuscript (Page 10, line 188-192) and included a negative control with

non-transgenic fish for FACS (Fig. 3c, d). In addition, according to the reviewer's suggestion, we have provided gene ontology analyses of *epd*⁺ cell-enriched genes in the revised manuscript (Supplementary Fig. 5; page 11, line 205-209) and updated the discussion with possible mechanisms underlying the regulation of muLECs by mLSCs and other possible functions of mLSCs (Page 18-21, line 362-371, 397-408).

Minor point:

5. In the introduction, the authors are referencing a manuscript claiming that muLECs are also present in mammals. This reviewer would caution the authors to be more hypothetical when translating the presence of muLECs to mammals as that paper is controversial and the presence of true muLECs in mammal remains extremely debatable.

Response : Because of the controversy, we have deleted this reference in the Introduction and Discussion of the revised manuscript.

6. This reviewer sees it as a missed opportunity to not hypothesize about the functional role of preventing muLECs to form tubular structure during normal development in the discussion section.

Response : We have followed the reviewer's suggestion to discuss potential roles of mLSCs in preventing muLECs to form tubular structure during normal development. Please see the Discussion section of the revised manuscript (Page 19, 20, line 382-390).

Reviewer #2 (Remarks to the Author):

Recent studies have shed light on the heterogeneous compositions of lymphatic endothelial cells (LECs) and their unique functions in specific tissues. One such population is the mural lymphatic endothelial cells (muLECs), which do not form vessel structures but remain dispersed, facilitating their clearance of metabolic waste from the brain. Despite the crucial functions of muLECs in brain immune homeostasis, how they form during development and how they maintain the unique morphology remain

largely unknown. In this manuscript, He et al. identified a distinct meningeal cell population which regulates the development, morphology maintenance, and regeneration of muLECs.

To begin with, the authors generated a specific reporter line for this cell population based on its specific expression of the gene *epd*. Using this reporter line, they comprehensively characterized the cellular and molecular features of this unique cell population. They demonstrated that these cells represent a previously unidentified population distinct from other mural cell types. They further discovered that the *epd*⁺ cells are closely associated with blood vessels and muLECs during development and possess a pro-lymphangiogenic transcription profile, suggesting their potential role in muLEC formation and function. To explore the function of the *epd*⁺ cells, the authors specifically ablated these cells at different developmental stages and conditions and examined their impact on muLECs. Remarkably, they found that the loss of the *epd*⁺ cells led to the blockade of muLEC formation during development and disrupted their morphology and regenerate at later stages. These findings demonstrate an essential role of the *epd*⁺ cells in supporting muLEC development and function. Hence, the authors named these cells as meningeal lymphatic supporting cells (mLSCs).

In sum, the authors conducted elegant studies on this novel and important cell population. The experiments were well-designed, and the results were clearly presented. The major conclusions are well-supported by the experimental data. Overall, it is a great piece of work and well-suited for publication in Nature Communications. I have only a few minor comments and some suggestions.

1, The *epd*⁺ cells appear to be ubiquitously distributed and form a membranous structure that covers the entire meninges. Yet a substantial portion of the *epd*⁺ cells do not seem to directly connect to muLECs, raising the possibility that these cells may have additional functions beyond supporting muLECs. It would be valuable if the authors could discuss this issue in the discussion.

Response: We have included relevant discussions in the Discussion section of the revised manuscript (Page 18, 19, line 362-371).

2, In the cell depletion assays, it seems that the epd+ cells could not be recovered even after the removal of Mtz. Does this suggest that these cells are maintained and expanded by self-proliferation during development? It would be valuable for the authors to discuss the potential origin and lineage development of this cell population.

Response: After removal of Mtz, mLSCs were able to gradually recover in some larvae (Fig. 5m, 5p). Since the recovery of mLSCs varied across treatment conditions, the provided images in this study did not show similar degree of recovery. There are two single-cell sequencing data from two previous studies^{1,2}. One suggests that mLSCs originate from the mesoderm¹, and the other suggests that mLSCs belong to a subpopulation of mesenchyme².

3, In Fig. 3b, the sorting strategy for the target epd+ cells is not clearly depicted, and there is a lack of a fluorescence-negative control for gating.

Response: We have improved the statements of the sorting strategy for mLSCs (Page 10, line 188-192), and provided the corresponding fluorescence-negative control (Fig. 3a-d).

4, Line 51 and 66: the full name of muLECs and mLSCs should be provided when it is first mentioned in the main text.

Response: We have provided the full name of muLECs and mLSCs when first mentioned in the main text (Page 4, line 54, 68, 69).

References:

1. Farnsworth, D.R., Saunders, L.M., Miller, A.C. A single-cell transcriptome atlas for zebrafish development. *Dev Biol* **459**, 100-108 (2020).
2. Sur, A., Wang, Y., Capar, P., Margolin, G., Prochaska, M.K., Farrell, J.A. Single-cell analysis of shared signatures and transcriptional diversity during zebrafish development. *Dev Cell* **58**, 1–20 (2023).

REVIEWERS' COMMENTS

Reviewer #1 (Remarks to the Author):

First, I would like to thank the authors for a thorough and impressive response to my comments. The additions of quantifications are greatly strengthening the message of the manuscript. The addition of the re-analysis of the single cell sequencing also further validate the data the other generated.

I do not have further comments about this manuscript.

Reviewer #2 (Remarks to the Author):

The authors have addressed all my concerns in the revised manuscript, and I have not further question.